# Batch Training for Streaming Time Series: A Transferable Augmentation Framework to Combat Distribution Shifts

**Weiyang Zhang**                                                     *zhangweiyang12138@gmail.com*
*School of Computer Science and Technology, Harbin Institute of Technology (Shenzhen)*

**Xinyang Chen**[*]                                                     *chenxinyang95@gmail.com*
*School of Computer Science and Technology, Harbin Institute of Technology (Shenzhen)*

**Yu Sun**[*]                                                     *sunyu@nankai.edu.cn*
*College of Computer Science, DISSec, Nankai University*

**Weili Guan**                                                     *guanweili@hit.edu.cn*
*School of Information Science and Technology, Harbin Institute of Technology (Shenzhen)*

**Liqiang Nie**                                                     *nieliqiang@gmail.com*
*School of Computer Science and Technology, Harbin Institute of Technology (Shenzhen)*

**Reviewed on OpenReview:** *https://openreview.net/forum?id=Ht7rlkRCHq*

## Abstract

Multivariate time series forecasting, which predicts future dynamics by analyzing historical data, has become an essential tool in modern data analysis. With the development of deep models, batch-training based time series forecasting has made significant progress. However, in real-world applications, time series data is often collected incrementally in a streaming manner, with only a portion of the data available at each time step. As time progresses, distribution shifts in the data can occur, leading to a drastic decline in model performance. To address this challenge, online test-time adaptation and online time series forecasting have emerged as promising solutions. However, for the former, most online test-time adaptation methods are primarily designed for images and do not consider the specific characteristics of time series. As for the latter, online time series forecasting typically relies on updating the model with each newly collected sample individually, which may be problematic when the sample deviates significantly from the historical data distribution and contains noise, which may lead to a worse generalization performance. In this paper, we propose Batch Training with Transferable Online Augmentation (BTOA), which enhances model performance through three key ideas while enabling batch training. First, to fully leverage historical information, Transferable Historical Sample Selection (THSS) is proposed with theoretical guarantees to select historical samples that are the most similar to the test-time distribution. Then, to mitigate the negative impact of distribution shifts through batch training and take advantage of the unique characteristics of time series, Transferable Online Augmentation (TOA) is proposed to augment the selected historical samples from the perspective of amplitude and phase in the frequency domain in a two-stream manner. Finally, a prediction module that utilizes a series decomposition module and a two-stream forecaster is employed to extract the complex patterns in time series, boosting the prediction performance. Moreover, BTOA is a general approach that is readily pluggable into any existing batch-training based deep models. Comprehensive experiments under both ideal and practical experimental settings demonstrate that the proposed method exhibits superior performance across all seven benchmark datasets. Compared to state-of-the-art approaches, our method reduces the Mean Squared Error (MSE) by up to 13.7%.

---

[*]Corresponding authors.

# 1 Introduction

Time series forecasting is crucial in real-world applications and is widely used across various fields, such as weather forecasting (Zhang et al., 2022a), power demand prediction (Gasparin et al., 2022), traffic flow analysis (Jin et al., 2021), and financial market modeling (Lai et al., 2018). In these practical applications, time series forecasting techniques not only help decision-makers better plan and optimize resources but also improve system efficiency and stability, driving the intelligent development of various industries. To improve forecasting accuracy, recent research has proposed advanced forecasting methods (Zhou et al., 2022; Wu et al., 2021; 2022). However, they typically rely on a conventional machine learning assumption that the training and test data follow the same distribution. This assumption often does not hold in real-world applications, where dataset shifts frequently occur (Quionero-Candela et al., 2009). Consequently, model performance can be significantly degraded when tested with data that deviates substantially from the training distribution. It is also worth noting that due to the inherent temporal nature, time series often arrive continuously in real-world scenarios, which means that models are typically required to handle streaming data. Recently, online test-time adaptation and online time series forecasting have emerged as promising solutions to address this issue, allowing pre-trained models to adapt to previously unseen data distributions during inference without the need for labeled data (Wang et al., 2023; Liang et al., 2023).

Unlike traditional batch training methods, online test-time adaptation and online time series forecasting adapt models in real-time using streaming data. Current online test-time adaptation methods can be broadly classified into three categories (Liang et al., 2023): (1) Data-based methods (Gong et al., 2024; Wang et al., 2022a), which focus on maximizing prediction consistency across different test datasets. (2) Model-based methods (Jang et al., 2022; Liu et al., 2023; Shu et al., 2022a), which aim to modify the original model architecture by adapting specific layers or mechanisms. (3) Optimization-based methods (Wang et al., 2022b; Shu et al., 2022b; Mummadi et al., 2021), which focus on optimizing prediction results using various optimization techniques. However, most existing online test-time adaptation methods are predominantly designed for image-based tasks, with few approaches specifically tailored for the complex patterns inherent in time series data. Current online time series forecasting methods typically utilize traditional Bayesian theory or add additional adapter modules to achieve adaptation (Pham et al., 2022; Zhang et al., 2023). These methods often rely on updating the model individually with each newly collected sample. When a sample deviates significantly from the historical data distribution and may contain substantial noise, these approaches can lead to reduced generalization performance.

In this paper, Batch training with Transferable Online Augmentation (BTOA) framework is proposed for online test-time adaptation in time series forecasting, with the challenges of online test-time adaptation being addressed through three key innovations. Firstly, to fully leverage historical distribution information, we introduce the Transferable Historical Sample Selection (THSS) module with theoretical guarantees to precisely select historical samples from the memory bank that are closest to the test-time distribution. This overcomes the inefficiency of traditional online methods relying on random sampling or full-volume updates (Chaudhry et al., 2019; Buzzega et al., 2020), enabling intelligent activation and on-demand utilization of historical information. Secondly, to address distribution shifts, we propose the Transferable Online Augmentation (TOA) module, which enables batch training while avoiding the frequency-domain distortion (Verma et al., 2021; Demirel & Holz, 2024) caused by traditional time-domain data augmentation methods (e.g., Linear-Mixup (Zhang et al., 2017), Cut-Mixup (Yun et al., 2019)). TOA performs decoupling in the frequency domain and applies two-stream augmentation to the selected samples from both the amplitude and phase dimensions, fully preserving the amplitude-phase coupling information during the augmentation process. This approach uniquely maintains the critical frequency-domain characteristics that are essential for time series forecasting. Finally, a prediction block consisting of a series decomposition module and a two-stream forecaster generates predictions by achieving differential modeling of seasonal components and trend components in non-stationary time series. This design extracts complex patterns in time series data, significantly enhancing prediction performance by capturing hierarchical temporal dependencies.

Our main contributions are summarized as follows:

- The Batch training with Transferable Online Augmentation (BTOA) framework is proposed to address the distribution shift in online learning from three key perspectives. First, to fully leverage

historical distribution information, the Transferable Historical Sample Selection (THSS) module is introduced to select historical samples with minimal distribution discrepancy from the test-time distribution, enabling intelligent adaptation and on-demand activation of historical information. Second, the Transferable Online Augmentation (TOA) module is proposed to perform two-stream augmentation on the selected samples from the perspectives of frequency-domain amplitude and phase, effectively overcoming the frequency distortion limitations of traditional time-domain augmentation. This approach preserves essential frequency-domain features and enables batch training, thereby alleviating distribution shift. Finally, a prediction module is employed to achieve differential modeling, capturing complex temporal patterns and enhancing forecasting performance.

- BTOA is a general approach that is readily pluggable into any online time series forecasting model. This approach effectively mitigates the negative impact of noise in test-time samples, alleviates distribution shift, and enhances the effectiveness and robustness of the online learning model.
- We conducted experiments on seven popular real-world datasets in both ideal and practical scenarios. The results show that our method demonstrates excellent performance across all benchmark datasets. Compared to state-of-the-art Methods, our method reduces the Mean Squared Error by up to 13.7%.

## 2 Related work

### 2.1 Online Test-time Adaptation

Online test-time adaptation (OTTA) continuously updates the model in real-time as it encounters new data during inference. This ensures swift adaptation to evolving data distributions without altering the original training procedure (Chen et al., 2022; Nguyen et al., 2023; Zhang et al., 2022b). Notably, TENT (Wang et al., 2020) addresses distributional shift by dynamically adjusting batch normalization parameters through entropy loss minimization during inference. Similarly, EATA (Niu et al., 2022) introduces a selective approach to optimizing unsupervised surrogate losses akin to TENT, focusing solely on reliable and informative data points. ViDA (Liu et al., 2023) employs supervision of the student output by leveraging predictions from the teacher with augmented input. Additionally, it introduces high/low-rank adapters that are updated to accommodate continuous online test-time adaptation. ECL (Zeng et al., 2024) marks a departure from traditional methods by integrating a memory bank containing output distributions to establish thresholds for complementary labels. This innovative approach ensures the memory bank's continual relevance and effectiveness through periodic updates with the latest model parameters. Although these methods have shown promising results in the fields of computer vision and natural language processing (Wang et al., 2023; Liang et al., 2023), they do not take advantage of the unique characteristics of time series. Dish-TS (Fan et al., 2023) proposes the Dual-CONET framework, which learns the distribution discrepancies in input and output spaces respectively, and introduces a coefficient network to mitigate intra- and inter-space distribution differences. SOLID (Chen et al., 2024) employs a residual-based distribution shift detector to quantify the model's vulnerability to distribution shifts by evaluating the mutual information between prediction residuals and their corresponding contexts. TAFAS (Kim et al., 2025) flexibly adjusts the source forecaster to continually adapt to evolving test distributions while preserving the core semantic information learned during pre-training.

### 2.2 Online Time Series Forecasting

Online time forecasting focuses on streaming data, that is, for each $N$ variate sample $\mathbf{x}_i$ received, the model constructs an $L$-length look-back window $\mathbf{X}$ and outputs a $H$-length prediction window $\mathbf{Y}$, and then the true values are used to improve the model's performance in predicting the next sample. Online time series forecasting has a wide range of real-world applications due to the sequential nature of the data (Anava et al., 2013; Gultekin & Paisley, 2018; Aydore et al., 2019).

Previous methods have attempted to solve the online time series forecasting problem using Bayesian continuous learning theory, however, they are unable to quickly utilize information from historical samples. Inspired by Complementary Learning Systems (CLS) theory, FsNet (Pham et al., 2022) achieves great online time series forecasting by quickly adapting to historical data using the adapter module and slowly learning

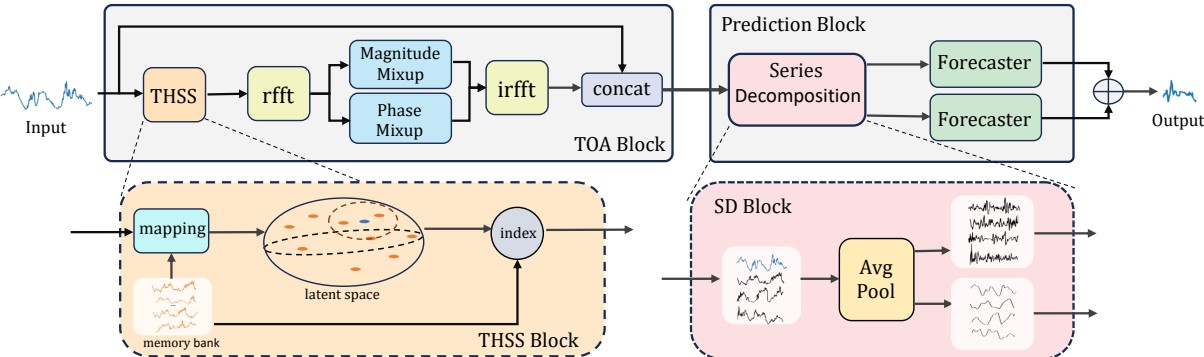

Figure 1: Overall architecture of Batch training with Transferable Online Augmentation (BTOA). The THSS block is used to select historical samples, the TOA block implements batch training through data augmentation, and the Prediction block generates the final output.

the newly collected sample with the Temporal Convolutional Network architecture. OneNet (Zhang et al., 2023) builds on FsNet by exploring the need for inter-channel dependencies, using the Online Convex Programming module to balance cross-time dependencies with cross-variate dependencies. This allows OneNet to achieve significant performance gains on some datasets with multiple variates such as the ECL dataset. Current models update based on a single received sample when processing streaming data. However, if a single sample is noisy, it can disrupt the optimal update path and significantly degrade model performance. To mitigate this issue, our BTOA implements batch training, which enhances the model's robustness against noisy data while maintaining effective online test-time adaptation. Proceed (Zhao & Shen, 2024) estimates inter-concept distribution shifts and uses an adaptive generator to effectively translate the estimated shifts into parameter adjustments, proactively adapting the model to test samples.

## 3 Method

**Problem Formulation.** Given a well-trained time series forecasting model $f$ train on the training set and a sequence of unlabeled time series segments. Distribution shift refers to the problem that arises when the distribution of the test data (target) diverges from that of the training data (source) (Liang et al., 2023). This phenomenon presents substantial challenges for machine learning systems deployed in practical scenarios (Saenko et al., 2010; Chen et al., 2017). Online test-time adaptation aims to leverage the labeled knowledge embedded in the prediction model $f$ to infer the future values of samples under distribution shift, in an online manner. In this problem, the learning process takes place over a sequence of rounds, where the model receives a $L$-length look-back window $\mathbf{X} = \{x_1, \ldots, x_L\} \in \mathbb{R}^{N \times L}$ and outputs the forecast window $\mathbf{Y} = \{y_1, \ldots, y_H\} \in \mathbb{R}^{N \times H}$. The true values $\hat{\mathbf{Y}}$ are then revealed to improve the model's performance in the next rounds. Our goal is to continuously optimize the prediction model $f$, which can mitigate the negative impact of distribution shifts.

**Structure overview.** Figure 1 illustrates the comprehensive workflow of Batch Training with Transferable Online Augmentation (BTOA). BTOA is meticulously structured into three principal modules: a transferable historical sample selection module that fully leverages historical distribution information, a transferable online augmentation module that enables batch training to alleviate the negative effects of distribution shift, and a prediction block that extracts complex temporal patterns and produces predictions.

### 3.1 Transferable Historical Sample Selection

The distribution of the data stream changes dynamically over time, which can adversely affect time series forecasting accuracy. To mitigate this issue, Transferable Historical Sample Selection is proposed to ef-

fectively utilize historical data. Firstly, we aim to select historical samples that have smaller distribution discrepancy to the test-time distribution.

Specifically, we establish a memory bank, which is a set and denoted as $\mathcal{M}$. $\mathcal{M}$ stores historical samples and is updated using a First-In-First-Out (FIFO) policy to maintain a fixed size. Upon receiving test-time sample, we use the THSS module to select historical samples that are semantically similar to the test-time sample and most closely align with test-time distribution. This selection is achieved through a mapping model, where intuitively, any model that can preserve semantic consistency between the original and mapped data can be used. Due to the inherent ability of the Variational Autoencoder (VAE) (Higgins et al., 2017) to maintain semantic consistency between input and output, we choose VAE model as our mapping model. Initially, the VAE model is pre-trained in an unsupervised manner on the training set, and during online test-time adaptation, its parameters are frozen to remain unchanged. Once the test-time sample $\mathbf{X}_{test}$ is introduced, both the test-time sample and the historical samples stored in $\mathcal{M}$ are projected into a latent space. We then calculate the distances between these samples in the latent space and select those historical samples that have smaller distribution discrepancy to the test-time distribution. These selected samples are semantically similar to the test-time sample, forming the selected historical sample set $\mathcal{X}_h$. The above procedure can be formulated as follows:

$$
\begin{aligned}
&\mathbf{z}_{test} = E(\mathbf{X}_{test}), \mathbf{z}_i = E(\mathbf{m}_i), \forall \mathbf{m}_i \in \mathcal{M} \\
&d_i = \text{cosine\_similarity}(\mathbf{z}_{test}, \mathbf{z}_i) = \frac{\mathbf{z}_{test} \cdot \mathbf{z}_i}{\|\mathbf{z}_{test}\| \|\mathbf{z}_i\|} \\
&\mathcal{X}_h = \{\mathbf{m}_i\}_{i \in S_n} \text{ where } S_n = \arg\text{sort}_i(d_i)[:n],
\end{aligned}
\tag{1}
$$

where $E(\cdot)$ represents the encoder of VAE model, and $n$ is a hyperparameter indicating the number of historical samples we need to utilize. By effectively utilizing historical samples that have a smaller distribution discrepancy to test-time distribution, we alleviate distribution shift and boost prediction performance.

## 3.2 Transferable Online Augmentation

We mitigate the negative impact of distribution shift during online test-time adaptation by introducing batch training, which can be achieved through data augmentation techniques. However, existing data augmentation methods, such as Linear-Mixup (Zhang et al., 2017) and Cut-Mixup (Yun et al., 2019), primarily mix time series in the time domain, which can affect the frequency domain information that is crucial for accurate prediction (Demirel & Holz, 2024; Ullrich et al., 2020; Zhang et al., 2022c). Since distribution shift is more pronounced in online test-time adaptation, preserving the frequency domain information of time series becomes particularly important.

To preserve the frequency domain information, we propose a two-stream augmentation approach that focuses on both the amplitude and phase in the frequency domain, and we select the aforementioned set of selected historical samples $\mathcal{X}_h$, which are closer to the test-time distribution, as the source for augmentation. By doing so, we ensure that the augmented instances' phase and amplitude are properly interpolated based on the test-time sample, avoiding destructive interference in the frequency domain. We first apply the Fast Fourier Transform (FFT) to both the test-time sample and historical samples stored in the set $\mathcal{X}_h$, decomposing them into amplitude and phase components, which can be formulated as:

$$
\text{A}(\mathbf{X}_i)e^{j\text{P}(\mathbf{X}_i)} = \mathcal{F}(\mathbf{X}_i), \quad \mathbf{X}_i \in \{\mathbf{X}_{test}\} \cup \mathcal{X}_h,
\tag{2}
$$

where $\mathcal{F}$ denotes the Fast Fourier Transform, $\text{A}(\cdot), \text{P}(\cdot)$ means the amplitude and phase. Then, we combine the amplitude and phase of the test-time sample with those of the historical samples. This process ensures that the frequency domain information remains intact while enhancing the data with relevant historical patterns. To ensure more appropriate historical samples, we perform aggressive data augmentation primarily using historical samples when their distance from the test-time sample in the latent space is small, indicating similar distributions. Conversely, when the distance between the historical and test-time sample is large, implying a significant distributional shift, we prioritize the test-time sample for data augmentation.

Specifically, the process of mixup is:

$$\mathrm{A}(\mathcal{X}_{aug}) = \{\mathbf{X}_j | \lambda_A \mathrm{A}(\mathbf{X}_{test}) + (1 - \lambda_A)\mathrm{A}(\mathbf{X}_j), \mathbf{X}_j \in \mathcal{X}_h\} \tag{3}$$

$$\mathrm{P}(\mathcal{X}_{aug}) = \{\mathbf{X}_j | \lambda_P \mathrm{P}(\mathbf{X}_{test}) + (1 - \lambda_P)\mathrm{P}(\mathbf{X}_j), \mathbf{X}_j \in \mathcal{X}_h\}, \tag{4}$$

where $\mathcal{X}_{aug}$ means the augmented sample set. $\lambda_A, \lambda_P$ are hyperparameters representing the mixing coefficients for amplitude and phase, respectively. When the distance between latent vectors is below a distance threshold, we sample the mixing coefficients for amplitude and phase from a uniform distribution, denoted as $\lambda_A, \lambda_P \sim \mathcal{U}(\beta, 1.0)$, prioritizing data augmentation on the historical samples, with $\beta$ being a lower value. Conversely, if the distance exceeds the threshold, we focus on augmenting the test-time sample. In this case, the coefficients are drawn from a truncated normal distribution, $\lambda_A, \lambda_P \sim \mathcal{N}(\mu, \theta)$, characterized by a high mean and low standard deviation. The process for determining the sampling distribution of $\lambda_A$ and $\lambda_P$ is as follows:

$$\lambda_A, \lambda_P \in \begin{cases} \mathcal{U}(\beta, 1.0), & \text{if } d_i \leq \tau \\ \mathcal{N}(\mu, \theta), & \text{if } d_i > \tau, \end{cases} \tag{5}$$

where $d_i$ represents the distance between latent vectors, and $\tau$ denotes a predefined distance threshold. Finally, the augmented sample set $\mathcal{X}_{aug}$ is obtained by applying the inverse FFT to the mixed components. For ease of understanding, we use $\mathbf{X}_{aug}$ to represent an element of $\mathcal{X}_{aug}$ hereafter. As shown below:

$$\mathbf{X}_{aug} = \mathcal{F}^{-1}\left(A(\mathbf{X}_{aug})e^{jP(\mathbf{X}_{aug})}\right), \tag{6}$$

where $\mathcal{F}^{-1}$ denotes the inverse Fast Fourier Transform. After obtaining the augmented set $\mathbf{X}_{aug}$, we concatenate these augmented samples with the test-time sample and input them as a batch into the next module for training. Compared to the previous approach, which only used the test-time sample to update the model, this method significantly reduces the negative impact of noise in the test-time sample on model optimization.

### 3.3 Prediction Model and Training Objective

**Prediction Model.** To effectively learn complex temporal patterns in time series forecasting, we use series decomposition (RB, 1990; Anderson, 1976). This technique simplifies complex raw data, allowing the model to make better predictions. Specifically, we extract the trend component of the time series by applying a moving average kernel to the input series. The difference between the trend component and the original series is regarded as the seasonal component. These components reflect the long-term trend and cyclical relationship of the time series, respectively. The series decomposition is handled as follows:

$$\mathbf{X}_t = \text{AvgPool}(\text{padding}(\mathbf{X}_{aug})), \tag{7}$$

$$\mathbf{X}_s = \mathbf{X}_{aug} - \mathbf{X}_t, \tag{8}$$

where $\mathbf{X}_t, \mathbf{X}_s$ denote the extracted long-term trend and seasonal terms, respectively. We use padding to maintain the original series length, and then apply the AvgPool layer for moving average calculations.

After decomposition, the trend component $\mathbf{X}_t$ and the seasonal component $\mathbf{X}_s$ will be fed into two-stream forecaster with identical structures. The outputs from two-stream forecaster are combined to generate the final prediction $\mathbf{Y}$. As shown below:

$$\mathbf{Y} = \text{Forecaster}_s(\mathbf{X}_s) + \text{Forecaster}_t(\mathbf{X}_t). \tag{9}$$

**Training Objective.** We use the $L2$ loss to optimize the parameters of the BTOA model, with the loss function defined as:

$$\mathcal{L} = \frac{1}{N}\sum_{j=1}^{N}\left\|\hat{\mathbf{Y}}_{1:H}^j - \mathbf{Y}_{1:H}^j\right\|, \tag{10}$$

where $N$ represents the number of channels in the time series. During the test-time adaptation process, we use the mean MSE and MAE between the ground truth $\hat{\mathbf{Y}}$ and the model's predicted output $\mathbf{Y}$ across all samples as the final evaluation metrics to compare model performance.

It is worth noting that BTOA is a general module designed to mitigate the distributional shift that occur during the learning process. This adaptability makes it applicable to any online time series forecasting model. The inherent flexibility of BTOA allows it to be integrated seamlessly with a variety of models, enhancing their robustness against changes in data distribution. Moreover, BTOA is not limited to a specific algorithm or framework. This means that as more advanced deep models are developed, BTOA can be incorporated into these advanced deep models to further improve performance.

### 3.4 Theoretical Insights

In the **Transferable Historical Sample Selection** module, we need to select a mapping model that can reflect the semantic consistency of samples before and after mapping. Proposition 3.1 theoretically demonstrates the rationale for using the VAE model as a mapping model.

**Proposition 3.1** *(**Consistency in Latent Space** (Li et al., 2022)) Given a well-trained unconditional VAE with the encoder $E(\cdot)$ that produces distribution $p_E(z|\mathbf{x})$, the decoder $D(\cdot)$ that produces distribution $q_D(\mathbf{x}|z)$ while the prior for $z$ is $p(z)$, let $\mathbf{z_1}$ and $\mathbf{z_2}$ be two latent vectors of two different real samples $\mathbf{x_1}$ and $\mathbf{x_2}$, i.e., $E(\mathbf{x_1}) = \mathbf{z_1}$ and $E(\mathbf{x_2}) = \mathbf{z_2}$. If the distance $d(\mathbf{z_1}, \mathbf{z_2}) \leq \delta$, then $D(\mathbf{z_1})$ and $D(\mathbf{z_2})$ will have a similar semantic label as in Equation equation 11.*

$$|I(D(\mathbf{z_1}); \mathbf{y}) - I(D(\mathbf{z_2}); \mathbf{y})| \leq \epsilon, \tag{11}$$

*where $\epsilon$ stands for tolerable semantic difference, $\delta$ is the maximum distance to maintain semantic consistency, and $d(\cdot)$ is a distance measure such as cosine similarity between two vectors.*

Let the historical sample set $\mathcal{P}$ be the set that includes the training set and all samples received prior to the test-time sample. The test-time sample set $\mathcal{Q}$ refers to the samples being received at present. Due to distribution shift, the data distributions of $\mathcal{P}$ and $\mathcal{Q}$ may differ. In the **Transferable online augmentation** module, the augmented set $\mathcal{X}_{aug}$ derived from the selected historical sample set $\mathcal{X}_h$, has a data distribution that is closer to the historical sample set $\mathcal{P}$ compared to using the test-time sample alone. Proposition 3.2 provides theoretical support for the performance advantages of using $\mathcal{X}_{aug}$ as input, suggesting that it can lead to a smaller upper bound on the generalization error. Moreover, our choice of $L2$ loss as the loss function aligns with the requirements of the proposition.

**Proposition 3.2** *(**Generalization Error Upper Bound** (Mansour et al., 2009)) Let $f_Q^* \in \arg\min_{f \in F} \mathcal{L}_{\mathcal{Q}}(f, G_Q)$ and similarly let $f_P^*$ be a minimizer of $\mathcal{L}_{\mathcal{P}}(f, G_P)$. Note that these minimizers may not be unique. For adaptation to succeed, it is natural to assume that the average loss $\mathcal{L}_{\mathcal{Q}}(f_Q^*, f_P^*)$ between the best-in-class hypotheses is small. Under that assumption and for a small discrepancy distance, there is a useful bound on the error of a hypothesis with respect to the test-time sample set as in Equation equation 12.*

$$\mathcal{L}_{\mathcal{Q}}(f, G_Q) \leq \mathcal{L}_{\mathcal{Q}}(f_Q^*, G_Q) + \mathcal{L}_{\mathcal{P}}(f, f_P^*) + disc_L(\mathcal{Q}, \mathcal{P})$$
$$+ \min\{\mathcal{L}_{\mathcal{P}}(f_P^*, f_Q^*), \mathcal{L}_{\mathcal{Q}}(f_P^*, f_Q^*)\}, \tag{12}$$

*where $G$ represents the ideal prediction model and the loss function $\mathcal{L}$ is symmetric and obeys the triangle inequality.*

## 4 Experiments

In this section, we evaluated BTOA across a range of online time series forecasting applications, demonstrating its effectiveness in diverse scenarios. In addition to the primary evaluation, we conducted comprehensive ablation studies to investigate the contribution of each individual component of BTOA.

Table 1: Statistics of popular datasets for benchmark.

| Datasets | ETTh1 | ETTh2 | ETTm1 | ETTm2 | Weather | Electricity | Traffic |
|---|---|---|---|---|---|---|---|
| Features | 7 | 7 | 7 | 7 | 21 | 321 | 862 |
| Timesteps | 17420 | 17420 | 69680 | 69680 | 52695 | 26304 | 17544 |
| ADF | -5.90 | -4.13 | -14.98 | -5.66 | -26.66 | -8.44 | -15.02 |

**Dataset.** We evaluate the performance of BTOA on seven real-world datasets, including ETT (with 4 subsets), Weather, ECL, Traffic used in iTransformer (Liu et al., 2024). The presence and severity of distributional shifts in these datasets can be measured using the Augmented Dickey-Fuller (ADF) test statistic (Liu et al., 2022). Basic information about these datasets is provided in Table 1, where it can be observed that they exhibit varying degrees of distributional shift. Detailed dataset descriptions are available in appendix A.1.

Table 2: Full results of the online time-series forecasting task. We compare extensive competitive models under different prediction lengths following the setting of Proceed. The best results are in **bold**, and the second best are underlined.

| Models | | BTOA | | Proceed | | SOLID++ | | OneNet | | FsNet | | CycleNet | | Dish-TS | | PatchTST | | DER++ | | ER | |
|---|---|---|---|---|---|---|---|---|---|---|---|---|---|---|---|---|---|---|---|---|---|---|
| Metric | | MSE | MAE | MSE | MAE | MSE | MAE | MSE | MAE | MSE | MAE | MSE | MAE | MSE | MAE | MSE | MAE | MSE | MAE | MSE | MAE |
| ETTh1 | 24 | **0.708** | **0.533** | 0.729 | 0.534 | 0.745 | 0.552 | 0.780 | 0.559 | 0.993 | 0.624 | 0.753 | 0.556 | 0.761 | 0.562 | 0.756 | 0.552 | 0.834 | 0.604 | 0.811 | 0.593 |
| | 48 | **0.806** | **0.576** | 0.886 | 0.593 | 0.848 | 0.593 | 0.896 | 0.600 | 1.089 | 0.664 | 0.857 | 0.596 | 0.882 | 0.603 | 0.887 | 0.601 | 0.921 | 0.635 | 0.901 | 0.627 |
| | 96 | **0.930** | **0.625** | 1.003 | 0.650 | 0.977 | 0.645 | 1.025 | 0.648 | 1.359 | 0.752 | 0.986 | 0.644 | 1.074 | 0.662 | 1.086 | 0.664 | 1.036 | 0.675 | 1.019 | 0.668 |
| ETTh2 | 24 | **1.688** | **0.563** | 1.801 | 0.603 | 2.021 | 0.609 | 2.606 | 0.638 | 2.941 | 0.696 | 2.228 | 0.636 | 1.931 | 0.612 | 1.893 | 0.608 | 2.790 | 0.714 | 2.492 | 0.684 |
| | 48 | **2.772** | **0.666** | 3.291 | 0.733 | 3.442 | 0.718 | 3.921 | 0.729 | 4.090 | 0.797 | 3.701 | 0.752 | 3.391 | 0.751 | 3.283 | 0.720 | 4.090 | 0.801 | 3.799 | 0.778 |
| | 96 | **4.880** | **0.800** | 5.790 | 0.847 | 5.802 | 0.868 | 6.248 | 0.927 | 6.245 | 0.945 | 6.674 | 0.920 | 6.054 | 0.911 | 5.976 | 0.887 | 6.363 | 0.925 | 6.022 | 0.905 |
| ETTm1 | 24 | 0.454 | 0.408 | **0.422** | **0.393** | 0.455 | 0.406 | 0.766 | 0.487 | 0.627 | 0.484 | 0.604 | 0.481 | 0.459 | 0.412 | 0.470 | 0.415 | 0.775 | 0.579 | 0.745 | 0.566 |
| | 48 | 0.592 | 0.477 | 0.579 | 0.462 | **0.578** | **0.461** | 0.978 | 0.582 | 0.855 | 0.560 | 0.808 | 0.564 | 0.608 | 0.481 | 0.598 | 0.474 | 0.847 | 0.605 | 0.817 | 0.592 |
| | 96 | **0.657** | 0.517 | 0.660 | 0.519 | 0.659 | **0.503** | 0.882 | 0.586 | 1.348 | 0.606 | 0.876 | 0.594 | 0.697 | 0.521 | 0.673 | 0.510 | 0.887 | 0.619 | 0.859 | 0.608 |
| ETTm2 | 24 | **0.614** | **0.406** | 0.617 | 0.409 | 0.699 | 0.420 | 0.768 | 0.435 | 0.877 | 0.449 | 0.895 | 0.442 | 0.693 | 0.427 | 0.658 | 0.415 | 1.838 | 0.619 | 1.626 | 0.589 |
| | 48 | **1.020** | 0.515 | 1.090 | 0.521 | 1.113 | **0.490** | 1.202 | 0.511 | 1.261 | 0.519 | 1.431 | 0.516 | 1.118 | 0.499 | 1.063 | 0.486 | 2.223 | 0.658 | 1.989 | 0.629 |
| | 96 | **1.556** | 0.577 | 1.979 | 0.586 | 1.822 | **0.563** | 3.102 | 0.621 | 4.224 | 0.736 | 2.121 | 0.585 | 2.041 | 0.603 | 1.867 | 0.565 | 2.733 | 0.700 | 2.505 | 0.674 |
| WTH | 24 | **0.712** | **0.352** | 0.728 | 0.366 | 0.735 | 0.372 | 0.774 | 0.385 | 0.877 | 0.404 | 0.916 | 0.463 | 0.741 | 0.372 | 0.732 | 0.368 | 1.502 | 0.691 | 1.444 | 0.668 |
| | 48 | 0.976 | **0.473** | **0.973** | 0.477 | 0.980 | 0.482 | 1.047 | 0.497 | 1.328 | 0.557 | 0.874 | 0.438 | 0.989 | 0.484 | 0.981 | 0.482 | 1.658 | 0.739 | 1.605 | 0.719 |
| | 96 | **1.261** | **0.584** | 1.264 | 0.592 | 1.263 | 0.597 | 1.320 | 0.603 | 1.714 | 0.679 | 1.188 | 0.560 | 1.269 | 0.601 | 1.263 | 0.592 | 1.793 | 0.780 | 1.750 | 0.764 |
| ECL | 24 | **3.941** | **0.281** | 3.978 | 0.285 | 4.156 | 0.294 | 4.112 | 0.293 | 6.194 | 0.381 | 4.343 | 0.299 | 4.432 | 0.299 | 4.143 | 0.294 | 11.877 | 0.583 | 11.304 | 0.566 |
| | 48 | **4.656** | **0.309** | 4.664 | 0.312 | 4.780 | 0.315 | 4.750 | 0.313 | 9.380 | 0.435 | 5.186 | 0.322 | 4.901 | 0.324 | 4.762 | 0.315 | 12.683 | 0.600 | 12.076 | 0.585 |
| | 96 | 5.675 | 0.348 | 5.672 | **0.340** | 5.835 | 0.340 | 5.703 | 0.336 | 12.851 | 0.464 | 6.350 | 0.349 | 6.791 | 0.371 | 5.791 | 0.341 | 13.221 | 0.601 | 12.671 | 0.585 |
| Traffic | 24 | **0.332** | **0.231** | 0.335 | 0.232 | 0.376 | 0.276 | 0.351 | 0.250 | 0.452 | 0.316 | 0.371 | 0.264 | 0.371 | 0.274 | 0.376 | 0.276 | 0.461 | 0.322 | 0.477 | 0.331 |
| | 48 | **0.356** | **0.243** | 0.358 | 0.245 | 0.378 | 0.244 | 0.374 | 0.262 | 0.498 | 0.337 | 0.395 | 0.278 | 0.393 | 0.269 | 0.380 | 0.267 | 0.501 | 0.347 | 0.519 | 0.353 |
| | 96 | **0.371** | **0.248** | 0.375 | 0.249 | 0.397 | 0.249 | 0.386 | 0.263 | 0.565 | 0.371 | 0.410 | 0.284 | 0.405 | 0.283 | 0.398 | 0.278 | 0.573 | 0.372 | 0.584 | 0.376 |
| AVG | | **1.664** | **0.463** | 1.771 | 0.473 | 1.812 | 0.476 | 1.999 | 0.501 | 2.798 | 0.556 | 2.025 | 0.511 | 1.905 | 0.491 | 1.811 | 0.481 | 3.314 | 0.627 | 3.143 | 0.612 |

**Baseline.** We evaluate multiple baseline methods in our experiments, covering approaches from continual learning, time series forecasting, and online learning. (1) Experience Replay and its variant DER++. **ER** stores historical data in a buffer and alternates it with newer samples during training. **DER++** (Buzzega et al., 2020) adds a knowledge distillation module compared to standard ER. (2) **Stationary** (Liu et al., 2022) focuses on modeling non-stationary time series through a De-stationary Attention module. (3) **Revin** (Kim et al., 2022) dynamically normalizes time series to mitigate the negative impact of non-stationarity, where we use PatchTST as a high-quality backbone. (4) **Dish-TS** (Fan et al., 2023) proposes a Dual-CONET framework that learns distribution discrepancies in input and output spaces respectively, using a

coefficient network to alleviate intra- and inter-space distribution differences. We also use PatchTST as a high-quality backbone here. (5) **iTransformer** (Liu et al., 2024) is a traditional time series forecasting method that models time series by transforming the roles of attention mechanisms and feed-forward networks. (6) **CycleNet** (Lin et al., 2024) uses a residual periodic prediction technique, leveraging learnable recursive periods to model inherent periodic patterns in sequences and making predictions on the modeled periodic residual components. (7) **FsNet** (Pham et al., 2022) avoids catastrophic forgetting of historical samples through TCN structures and adapter modules, enabling fast adaptation to new samples. (8) **OneNet** (Zhang et al., 2023) combines various strategies for time series forecasting. (9) **SOLID** (Chen et al., 2024) uses pre-trained parameters for linear probing at each update and does not inherit fine-tuned parameters from the previous update. We use the SOLID++ variant, continuously fine-tuning all model parameters on the online data. (10) **Proceed** (Zhao & Shen, 2024) estimates inter-concept distribution shifts and uses an adaptive generator to effectively translate the estimated shifts into parameter adjustments, proactively adapting the model to test samples.

Table 3: Full results of the online time-series forecasting task. We compare extensive competitive models under different prediction lengths following the setting of FsNet. The best results are in **bold**, and the second best are underlined.

| Models | | **BTOA** | | OneNet | | FsNet | | iTransformer | | CycleNet | | Dish-TS | | Revin | | Stationary | | DER++ | | DER | |
|---|---|---|---|---|---|---|---|---|---|---|---|---|---|---|---|---|---|---|---|---|---|
| Metric | | MSE | MAE | MSE | MAE | MSE | MAE | MSE | MAE | MSE | MAE | MSE | MAE | MSE | MAE | MSE | MAE | MSE | MAE | MSE | MAE |
| ETTh1 | 1 | **0.223** | 0.301 | 0.235 | 0.303 | 0.286 | 0.343 | **0.223** | **0.294** | 0.389 | 0.385 | 0.257 | 0.318 | 0.238 | 0.304 | 0.383 | 0.395 | 0.239 | 0.305 | 0.240 | 0.316 |
| ETTh1 | 24 | **0.271** | **0.325** | 0.400 | 0.442 | 0.411 | 0.436 | 0.703 | 0.524 | 0.959 | 0.621 | 0.692 | 0.517 | 0.672 | 0.510 | 0.759 | 0.565 | 0.648 | 0.534 | 0.673 | 0.547 |
| ETTh1 | 48 | **0.265** | **0.356** | 0.447 | 0.454 | 0.402 | 0.452 | 0.828 | 0.570 | 1.080 | 0.661 | 0.941 | 0.622 | 0.792 | 0.557 | 0.747 | 0.570 | 0.606 | 0.525 | 0.634 | 0.538 |
| ETTh2 | 1 | 0.390 | 0.362 | 0.383 | 0.351 | 0.467 | 0.371 | 0.418 | 0.352 | **0.559** | 0.384 | 0.514 | 0.362 | 0.383 | **0.344** | 0.770 | 0.383 | 0.508 | 0.375 | 0.508 | 0.376 |
| ETTh2 | 24 | **0.505** | **0.397** | 0.538 | 0.414 | 0.693 | 0.473 | 1.716 | 0.587 | 2.206 | 0.602 | 1.584 | 0.794 | 1.741 | 0.581 | 2.090 | 0.659 | 0.828 | 0.540 | 0.808 | 0.543 |
| ETTh2 | 48 | **0.587** | **0.436** | 0.604 | 0.445 | 0.867 | 0.516 | 2.781 | 0.676 | 3.254 | 0.692 | 2.119 | 0.677 | 2.762 | 0.664 | 2.938 | 0.722 | 1.157 | 0.577 | 1.136 | 0.571 |
| ETTm1 | 1 | 0.106 | 0.187 | 0.117 | 0.202 | **0.104** | 0.187 | 0.106 | 0.192 | 0.125 | **0.210** | 0.120 | 0.204 | 0.122 | 0.208 | 0.111 | 0.197 | 0.110 | 0.192 | 0.114 | 0.197 |
| ETTm1 | 24 | **0.114** | **0.222** | 0.134 | 0.243 | 0.137 | 0.249 | 1.663 | 0.692 | 0.816 | 0.535 | 0.427 | 0.471 | 1.531 | 0.704 | 0.536 | 0.449 | 0.196 | 0.326 | 0.202 | 0.333 |
| ETTm1 | 48 | **0.118** | **0.227** | 0.118 | 0.228 | 0.124 | 0.240 | 1.648 | 0.722 | 1.322 | 0.692 | 0.553 | 0.549 | 1.018 | 0.614 | 1.433 | 0.721 | 0.208 | 0.340 | 0.220 | 0.351 |
| ETTm2 | 1 | 0.174 | 0.226 | 0.191 | 0.233 | 0.179 | 0.229 | **0.168** | **0.221** | 0.173 | 0.224 | 0.321 | 0.271 | 0.173 | 0.226 | 0.194 | 0.228 | 0.190 | 0.231 | 0.191 | 0.233 |
| ETTm2 | 24 | **0.206** | **0.259** | 0.267 | 0.261 | 0.233 | 0.276 | 0.639 | 0.430 | 0.659 | 0.411 | 0.611 | 0.450 | 0.652 | 0.435 | 0.954 | 0.579 | 0.307 | 0.345 | 0.310 | 0.347 |
| ETTm2 | 48 | **0.204** | **0.267** | 0.273 | 0.284 | 0.299 | 0.313 | 0.987 | 0.502 | 1.067 | 0.480 | 0.906 | 0.507 | 1.083 | 0.502 | 1.209 | 0.592 | 0.329 | 0.359 | 0.331 | 0.363 |
| WTH | 1 | **0.156** | **0.197** | 0.158 | 0.201 | 0.161 | 0.215 | 0.160 | 0.205 | 0.169 | 0.210 | 0.156 | 0.195 | 0.165 | 0.211 | 0.152 | 0.196 | 0.208 | 0.235 | 0.180 | 0.244 |
| WTH | 24 | **0.161** | **0.241** | 0.189 | 0.273 | 0.189 | 0.276 | 0.375 | 0.399 | 0.388 | 0.406 | 0.340 | 0.382 | 0.370 | 0.394 | 0.428 | 0.446 | 0.270 | 0.351 | 0.293 | 0.356 |
| WTH | 48 | **0.173** | **0.255** | 0.197 | 0.278 | 0.223 | 0.303 | 0.472 | 0.467 | 0.478 | 0.468 | 0.412 | 0.440 | 0.453 | 0.452 | 0.487 | 0.484 | 0.294 | 0.359 | 0.297 | 0.363 |
| ECL | 1 | 2.430 | 0.266 | 2.590 | 0.258 | 3.317 | 0.542 | **1.897** | **0.218** | 2.274 | 0.229 | 2.966 | 0.288 | 3.873 | 0.331 | 2.613 | 0.508 | 2.657 | 0.421 | 2.579 | 0.506 |
| ECL | 24 | **2.493** | **0.346** | 2.700 | 0.366 | 6.071 | 1.024 | 4.009 | 0.313 | 6.585 | 0.368 | 3.236 | 0.523 | 3.469 | 0.579 | 8.996 | 1.035 | 9.265 | 1.066 | 9.327 | 1.057 |
| ECL | 48 | **2.423** | 0.462 | 3.261 | 0.400 | 7.234 | 1.089 | 4.787 | 0.346 | 7.276 | 0.393 | 5.941 | 0.355 | 6.583 | 0.379 | 4.987 | 0.789 | 9.009 | 1.048 | 9.685 | 1.074 |
| Traffic | 1 | **0.232** | **0.205** | 0.233 | 0.215 | 0.295 | 0.253 | 0.298 | 0.321 | 0.313 | 0.318 | 0.410 | 0.315 | 0.257 | 0.295 | 0.418 | 0.325 | 0.280 | 0.241 | 0.286 | 0.247 |
| Traffic | 24 | **0.310** | **0.261** | 0.348 | 0.269 | 0.360 | 0.287 | 1.187 | 0.498 | 1.006 | 0.508 | 0.913 | 0.508 | 1.097 | 0.502 | 1.275 | 0.575 | 0.384 | 0.289 | 0.383 | 0.299 |
| Traffic | 48 | **0.351** | **0.293** | 0.384 | 0.302 | 0.378 | 0.297 | 1.615 | 0.568 | 1.786 | 0.570 | 1.479 | 0.583 | 1.765 | 0.617 | 0.398 | 0.295 | 0.391 | 0.310 | 0.394 | 0.307 |
| AVG | | **0.567** | **0.285** | 0.656 | 0.306 | 1.068 | 0.395 | 1.183 | 0.418 | 1.630 | 0.454 | 0.805 | 0.456 | 1.452 | 0.434 | 1.320 | 0.504 | 1.325 | 0.425 | 1.371 | 0.437 |

**Implementation details.** To ensure a more fair comparison of the model's performance, we followed the experimental setups of FsNet (Pham et al., 2022) and Proceed (Zhao & Shen, 2024), testing the model under two different scenarios: one is the ideal experimental setup from FsNet, and the other is the practical experimental setup from Proceed. For the ideal experimental setup, we set the prediction lengths to 1, 24, and 48. For the practical experimental setup, we set the prediction lengths to 24, 48, and 96 to better compare the model's performance with longer prediction lengths. Since learning is conducted in sequential rounds, the model receives a look-back window in each round and predicts the forecast window. All models are evaluated using cumulative Mean Squared Error (MSE) and Mean Absolute Error (MAE), which assess the models' performance over the entire learning process. We use the AdamW (Loshchilov & Hutter, 2017)

optimizer to minimize the $L2$ loss. To simulate streaming data, we set the batch size to 1, reflecting the arrival of data in a streaming fashion. Our method is a general-purpose module. In the ideal setup, we instantiate it using OneNet as the forecaster, while in the practical setup, we consistently use PatchTST as the forecaster. More details about the implementation, architectures, and hyperparameters with the trained VAE model are given in Appendix B.4.

## 4.1 Online Forecasting Results

**Cumulative performance.** To validate the effectiveness of the BTOA framework, we compared its forecasting performance with other baselines under two experimental settings across seven popular datasets. Each experiment was repeated three times, and we report the average results. In Appendix A.3 and B.3, we provide the standard deviations of the methods and additional experimental results, offering a more comprehensive evaluation of the model performance and the robustness of our approach across multiple trials. As shown in Tables 2 and 3, BTOA achieved the best performance in most cases. Notably, BTOA significantly improved forecasting accuracy on datasets with strong non-stationarity, such as ETTh2 and ETTm2. This highlights BTOA's critical role in addressing distribution shift issues during online test-time adaptation. By effectively leveraging historical samples, BTOA mitigates distribution shifts, reduces the impact of potential noise in test-time samples on the model, and significantly enhances model robustness.

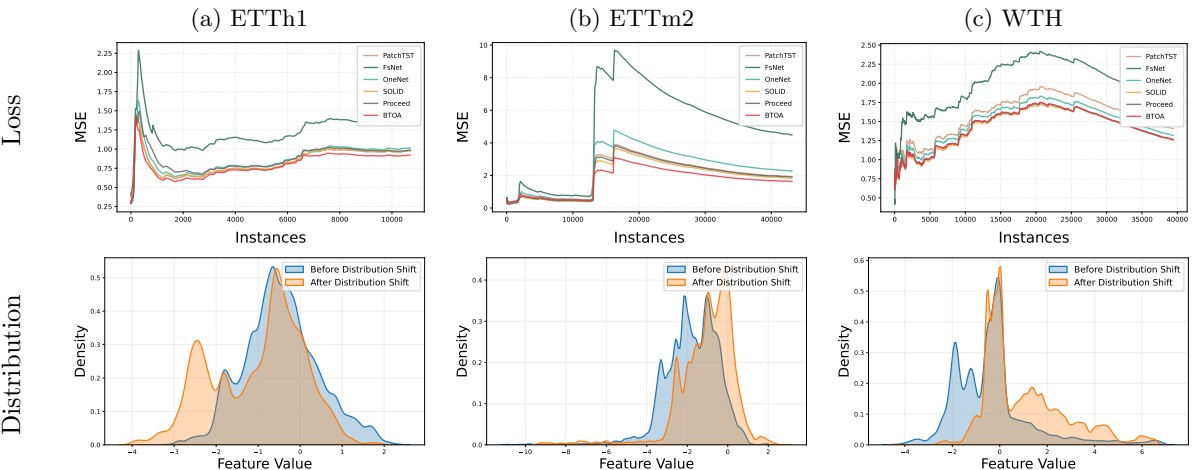

Figure 2: Evolution of the cumulative MSE loss and visualization of distribution shifts.

**Convergence of different deep models.** As shown in Figure 2, we analyze the convergence behavior of BTOA compared with baseline models across three datasets and select 200 time-point samples before and after loss peaks to observe their distribution differences. It is evident that loss curve peaks tend to occur in the early and middle stages of training, and samples before and after these peaks exhibit distribution shifts, fully indicating that changes in data distribution significantly impair model performance. Such distribution shifts pose particular challenges for time-series models, as they disrupt the patterns learned by the model, leading to temporary performance degradation. Traditional batch learning settings typically evaluate performance only on a small subset of data at the end of training, often neglecting the distribution shifts that occur during training, which results in suboptimal adaptation to practical scenarios. From Figure 2, we observe that when a distribution shift occurs, the increase in MSE for BTOA is significantly smaller than that of all baseline models. This demonstrates BTOA's superior capability to detect and rapidly adapt to distributional changes, thereby maintaining more stable performance. The key to this adaptability lies in the integration of its data augmentation module, which leverages historical data and a two-stream augmentation technique to capture shifts in both the amplitude and phase of the data, enhancing the model's robustness. This ability enables BTOA to make effective adjustments without compromising critical frequency domain information, sharply distinguishing it from other models that struggle with such changes. These results highlight BTOA's effectiveness in quickly adapting to distribution shifts, ensuring more stable and reliable performance. Its

capacity to mitigate the impact of such shifts underscores BTOA's potential as a versatile and robust solution for real-world time-series forecasting challenges.

Table 4: Performance metrics for different batch sizes.

| Dataset | ETTh1 | | | ETTm1 | | | WTH | | | ECL | | | Traffic | | |
|---|---|---|---|---|---|---|---|---|---|---|---|---|---|---|---|
| batch-size | 24 | 48 | 96 | 24 | 48 | 96 | 24 | 48 | 96 | 24 | 48 | 96 | 24 | 48 | 96 |
| 8 | 0.710 | 0.809 | 0.931 | 0.457 | 0.600 | 0.663 | 0.714 | 0.981 | 1.262 | 3.971 | 4.724 | 5.795 | 0.325 | 0.359 | 0.376 |
| 16 | **0.708** | 0.806 | **0.930** | **0.454** | **0.592** | **0.657** | **0.712** | **0.976** | **1.261** | **3.941** | **4.656** | **5.675** | 0.320 | **0.356** | **0.371** |
| 32 | 0.711 | **0.803** | 0.933 | 0.457 | 0.593 | 0.660 | 0.715 | 0.979 | 1.261 | 3.949 | 4.659 | 5.678 | **0.319** | 0.356 | 0.379 |
| 64 | 0.725 | 0.814 | 0.940 | 0.471 | 0.613 | 0.672 | 0.732 | 0.991 | 1.290 | 3.978 | 4.711 | 5.801 | 0.334 | 0.369 | 0.380 |

**The batch size at the online test-time adaptation.** In Table 4, we also investigated the impact of the batch size hyperparameter, which determines the number of historical samples to be augmented during the online batch training phase. Our experiments revealed that optimal performance is achieved with batch sizes of either 16 or 32. Considering both performance and computational efficiency, we selected a batch size of 16 as the optimal choice. This decision strikes a balance between ensuring high model performance and reducing training time, thus improving the model's practicality for real-world applications, especially when handling large-scale streaming data. By optimizing this hyperparameter, we can significantly accelerate the training process without sacrificing accuracy, making our approach better suited for time-sensitive tasks.

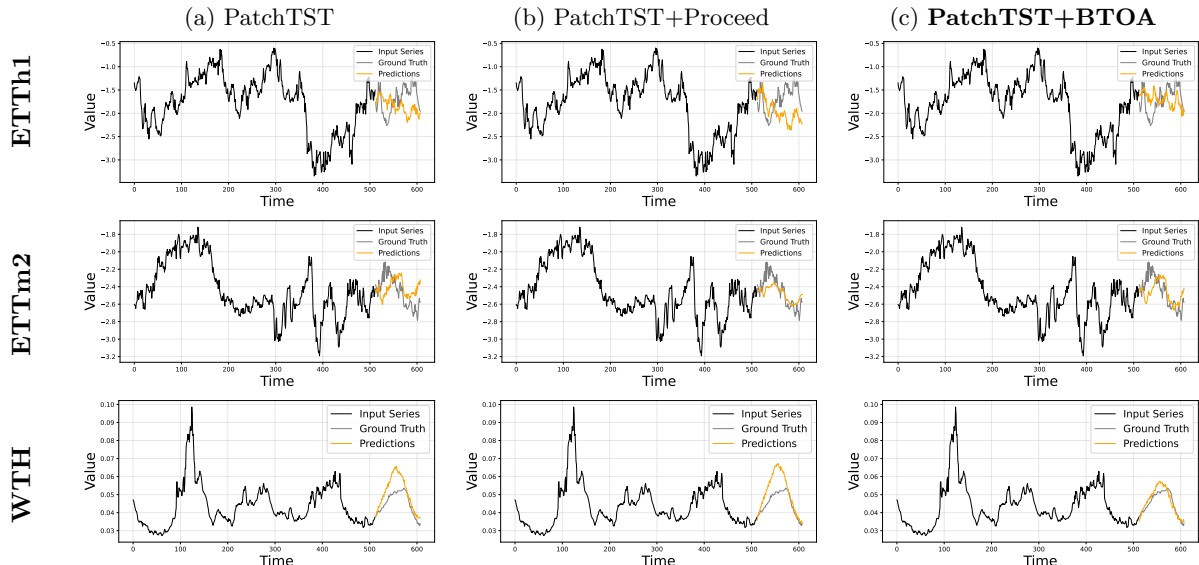

Figure 3: Forecast results under distribution shift with forecasting window $H = 96$.

**Visualization.** To validate the effectiveness of our method, we compare the prediction results of three models—PatchTST, Proceed, and our BTOA—on three datasets. Specifically, we selected samples from the post-peak loss curve period, as these samples better reflect the models' adaptability after distribution shifts occur. As shown in Figure 3, BTOA exhibits faster adaptation to distribution shifts and closer alignment with ground truth compared to baselines. Specifically, BTOA's predictions remain tightly aligned with true values, whereas PatchTST and Proceed show prolonged deviations. As training progresses, BTOA not only adapts faster but also captures complex temporal patterns more effectively, leading to superior prediction performance. This highlights BTOA's advantage in mitigating the negative impacts of distribution shifts and underscores its ability to maintain—and even improve—predictive accuracy over time, making it a more reliable solution for online scenarios.

## 4.2 Ablation Study

In this section, we will examine the contribution of each module in BTOA individually. The first module is Transferable Historical Sample Selection module, which selects historical samples that are close to the information of the test-time sample, a critical factor in fully leveraging historical information. The second module is the Transferable Online Augmentation module, which performs batch training to help mitigate noise present in the test-time sample and reduce distribution shifts. More ablation results are provided in Appendix B.2.

Table 5: Comparison of Historical Sample Selection Methods.

| Dataset | ETTh1 | | | ETTm1 | | | WTH | | | ECL | | | Traffic | | |
|---|---|---|---|---|---|---|---|---|---|---|---|---|---|---|---|
| Methods | 24 | 48 | 96 | 24 | 48 | 96 | 24 | 48 | 96 | 24 | 48 | 96 | 24 | 48 | 96 |
| closest-16 | 0.715 | 0.809 | 0.944 | 0.458 | 0.600 | 0.672 | 0.719 | 0.991 | 1.283 | 3.986 | 4.678 | 5.732 | 0.334 | 0.372 | 0.391 |
| L2-distance | 0.711 | 0.812 | 0.935 | 0.459 | 0.601 | 0.659 | 0.717 | 0.982 | 1.273 | 3.956 | 4.671 | 5.723 | 0.324 | 0.362 | 0.374 |
| THSS (ours) | **0.708** | **0.806** | **0.930** | **0.454** | **0.592** | **0.657** | **0.712** | **0.976** | **1.261** | **3.941** | **4.656** | **5.675** | **0.320** | **0.356** | **0.371** |

**The Components of BTOA.** First, we examine the THSS module. We conducted a comparison between three methods: (1) directly selecting the 16 most recent samples in time closest to the test-time sample for data augmentation, without choosing from the memory bank; (2) using L2 distance as a representative naive distance metric for the selection criterion; and (3) using the THSS module as the selection standard. As shown in Table 5, the THSS module outperforms the other two methods in selecting historical samples. This result demonstrates that the VAE successfully captures the semantic information of time series in the latent space, confirming the effectiveness of the latent space.

Table 6: Comparison of Data Augmentation Methods.

| Dataset | ETTh1 | | | ETTm1 | | | WTH | | | ECL | | | Traffic | | |
|---|---|---|---|---|---|---|---|---|---|---|---|---|---|---|---|
| Methods | 24 | 48 | 96 | 24 | 48 | 96 | 24 | 48 | 96 | 24 | 48 | 96 | 24 | 48 | 96 |
| Non-Aug | 0.756 | 0.882 | 1.086 | 0.470 | 0.598 | 0.673 | 0.732 | 0.981 | 1.263 | 4.143 | 4.762 | 5.791 | 0.376 | 0.380 | 0.398 |
| Linear-Mixup | 0.762 | 0.890 | 1.121 | 0.471 | 0.602 | 0.681 | 0.744 | 0.992 | 1.283 | 4.152 | 4.801 | 5.822 | 0.379 | 0.384 | 0.406 |
| Cut-Mixup | 0.753 | 0.869 | 0.941 | 0.467 | 0.608 | 0.669 | 0.729 | 0.983 | 1.271 | 4.097 | 4.722 | 5.731 | 0.355 | 0.374 | 0.380 |
| TOA (ours) | **0.708** | **0.806** | **0.930** | **0.454** | **0.592** | **0.657** | **0.712** | **0.976** | **1.261** | **3.941** | **4.656** | **5.675** | **0.320** | **0.356** | **0.371** |

Second, we evaluate the effectiveness of the TOA module. We compare four methods on five datasets: (1) without data augmentation; (2) using Linear-Mixup (Zhang et al., 2017); (3) using Cut-Mixup (Yun et al., 2019); and (4) using our proposed TOA module. Table 6 demonstrates the superior performance of the TOA method across all five datasets. This indicates that, for time series, simple time domain data augmentation can easily lead to interference between different series. In contrast, the TOA method performs data augmentation in the frequency domain, focusing on amplitude and phase, reducing the interference between time series and achieving positive data augmentation, thereby improving prediction performance.

Table 7: The Generality of BTOA.

| Dataset | ETTh1 | | | ETTm2 | | | WTH | | | ECL | | | Traffic | | |
|---|---|---|---|---|---|---|---|---|---|---|---|---|---|---|---|
| Methods | 24 | 48 | 96 | 24 | 48 | 96 | 24 | 48 | 96 | 24 | 48 | 96 | 24 | 48 | 96 |
| TCN | 1.006 | 1.138 | 1.360 | 0.899 | 1.233 | 1.823 | 0.872 | 1.418 | 1.805 | 6.964 | 8.155 | 10.001 | 0.478 | 0.530 | 0.573 |
| TCN+BTOA | **0.912** | **1.021** | **1.244** | **0.781** | **0.942** | **1.184** | **0.720** | **0.994** | **1.298** | **6.031** | **7.152** | **9.077** | **0.409** | **0.447** | **0.523** |
| iTransformer | 0.758 | 0.900 | 1.103 | 0.874 | 1.313 | 1.948 | 0.845 | 1.124 | 1.411 | 3.940 | 4.667 | 5.844 | 0.340 | 0.361 | 0.379 |
| iTransformer+BTOA | **0.702** | **0.835** | **0.991** | **0.802** | **1.302** | **1.766** | **0.733** | **0.998** | **1.280** | **3.818** | **4.596** | **5.705** | **0.329** | **0.351** | **0.376** |

**Generalisability.** As a plug-and-play module, BTOA has demonstrated significant effectiveness in improving model performance. In Table 7, we compare the results of adding BTOA to the TCN and iTransformer

models. The experimental results show that, in all models, the performance after adding BTOA outperforms the original versions of these models. These results fully validates the effectiveness of BTOA in enhancing the models' generalization ability and overall performance. Specifically, BTOA improves the model's robustness against performance degradation caused by distribution shifts in online time series prediction through batch-training. Additionally, the inclusion of BTOA also positively impacts the stability of model training. Therefore, BTOA, as a general-purpose module, not only improves the performance of various models but also enhances their feasibility and stability in real-world applications.

## 5    Conclusion

In this paper, a general online test-time adaptation method, Batch training with Transferable Online Augmentation (BTOA) for time series, is designed to effectively mitigate the performance degradation caused by distribution shifts during the test process. Our approach incorporates a transferable historical sample selection module, a transferable online augmentation module, and a prediction block. The THSS module fully leverages the distributional information from historical samples. Through the TOA module, we reintroduce batch training in the online setting to alleviate the negative impact of distribution shifts. Finally, the prediction block extracts complex patterns in time series, enabling more accurate outputs. Extensive experiments demonstrate that our approach can be integrated into any online time series forecasting model and achieves superior performance.

### Broader Impact Statement

Our work proposes a plug-and-play module from the perspective of online test-time adaptation to mitigate the negative impact of distribution shift. This is crucial for practical time series forecasting tasks. This research can serve as a reference for future studies in machine learning and data science, promoting the development of more complex and accurate online time series forecasting techniques. Therefore, our paper primarily focuses on scientific research and does not have any obvious negative social impact.

### Acknowledgements

This work was supported by the Shenzhen Science and Technology Program (ZDSYS20230626091203008), National Natural Science Foundation of China (62306085, 62302241, 62476071, 62236003, U23B2055, U24A20328), Shenzhen College Stability Support Plan (GXWD20231130151329002, GXWD20220817144428005).

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

## A  Additional experimental details

### A.1  Datasets

We conduct experiments on 7 real-world datasets to evaluate the performance of our method including (1) ETT(Electricity Transformer Temperature) (Wu et al., 2021) are collected from two electricity transformers with 7 factors. There are four subsets where ETTh1 and ETTh2 are recorded every hour, and ETTm1 and ETTm2 are recorded every 15 minutes. (2) WTH (Wu et al., 2021) includes 21 meteorological factors collected every 10 minutes from the Weather Station of the Max Planck Biogeochemistry Institute in 2020. (3) ECL (Wu et al., 2021) collects 321 customers' hourly electricity consumption. (4) Traffic (Wu et al., 2021)collects the road occupancy rates from different sensors on San Francisco freeways.

### A.2  Implementation Details

All experiments are implemented in PyTorch and conducted on a single NVIDIA RTX4090 24GB GPU. The learning rate for the experiments is set between 1e-3 and 7e-3.

### A.3  Details about the standard deviation

We report the standard deviations of BTOA for three backbones on seven datasets across three random seeds in Table 8.

Table 8: Standard deviation of three backbones.

| Backbones | Metric | ETTh1 | ETTh2 | ETTm1 | ETTm2 | Weather | Electricity | Traffic |
|---|---|---|---|---|---|---|---|---|
| TCN | MSE | 0.007 | 0.008 | 0.005 | 0.003 | 0.001 | 0.04 | 0.0007 |
| | MAE | 0.002 | 0.003 | 0.001 | 0.0005 | 0.0007 | 0.0002 | 0.001 |
| iTransformer | MSE | 0.001 | 0.01 | 0.01 | 0.001 | 0.009 | 0.04 | 0.0006 |
| | MAE | 0.0005 | 0.002 | 0.001 | 0.0001 | 0.004 | 0.0005 | 0.0005 |
| PatchTST | MSE | 0.001 | 0.03 | 0.008 | 0.003 | 0.001 | 0.04 | 0.0007 |
| | MAE | 0.0002 | 0.003 | 0.002 | 0.0003 | 0.0001 | 0.001 | 0.001 |

## B  Additional experimental Results

### B.1  Analysis of Inference Time and Memory Consumption

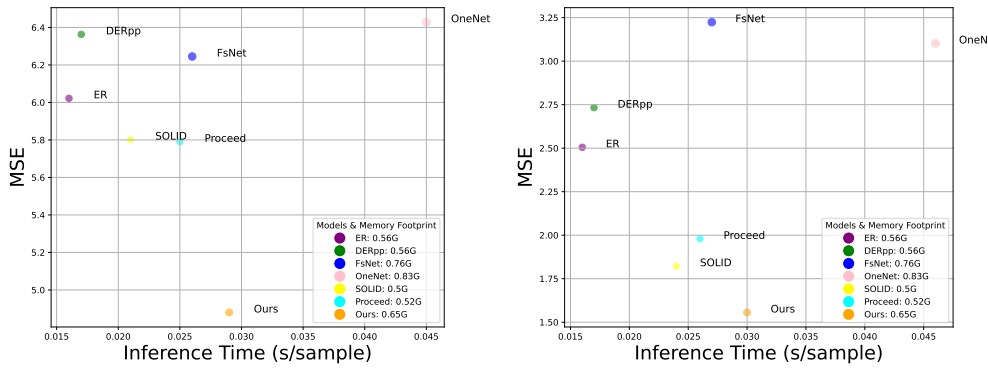

Figure 4: Comparison of inference time and memory consumption on ETTh2 and ETTm2.

Since online test-time adaptation deals with streaming data that arrives in a continuous and sequential manner, Figure 4 compares the inference time for a single sample and overall memory usage between our

method and the baselines. The results show that BTOA introduces minimal increases in inference time and memory usage, which are acceptable in an online setting, while outperforming the baseline methods. Notably, for the VAE module during inference, BTOA only requires simple forward propagation without the need for backpropagation. On the ETTh2 dataset, compared to Proceed, BTOA improves prediction performance by 18% with only a 9% increase in inference time; compared to OneNet, BTOA enhances prediction performance by 28% while reducing inference time by 35%. This comparison further highlights the superiority of BTOA.

## B.2 Ablation Study

Table 9: Performance Impact of Series Decomposition Block.

| Dataset | ETTh1 | | | ETTm1 | | | WTH | | | ECL | | | Traffic | | |
|---|---|---|---|---|---|---|---|---|---|---|---|---|---|---|---|
| Methods | 24 | 48 | 96 | 24 | 48 | 96 | 24 | 48 | 96 | 24 | 48 | 96 | 24 | 48 | 96 |
| Non-SD | 0.712 | 0.820 | 0.955 | 0.461 | 0.596 | 0.668 | 0.721 | 0.982 | 1.270 | 4.021 | 4.712 | 5.721 | 0.342 | 0.367 | 0.380 |
| SD | **0.708** | **0.806** | **0.930** | **0.454** | **0.592** | **0.657** | **0.712** | **0.976** | **1.261** | **3.941** | **4.656** | **5.675** | **0.320** | **0.356** | **0.371** |

we evaluate the effect of using a series decomposition module in the prediction block by comparing the performance of models with and without series decomposition. Although OneNet suggests that series decomposition is not a universally effective method for improving performance, Table 9 shows that for the BTOA model, series decomposition can effectively extract complex temporal patterns and enhance the model's predictive ability.

Table 10: Performance Impact of different distance threshold.

| Dataset | ETTh1 | | | WTH | | | Traffic | | |
|---|---|---|---|---|---|---|---|---|---|
| distance | 24 | 48 | 96 | 24 | 48 | 96 | 24 | 48 | 96 |
| 0.7 | 0.241 | 0.295 | 0.259 | 0.157 | 0.169 | **0.171** | 0.249 | **0.306** | 0.362 |
| 0.8 | **0.223** | **0.271** | **0.265** | **0.156** | **0.161** | 0.174 | **0.232** | 0.310 | **0.351** |
| 0.9 | 0.233 | 0.272 | 0.266 | 0.159 | 0.170 | 0.182 | 0.262 | 0.316 | 0.355 |

We also conduct ablation experiments regarding the distance threshold hyperparameter as shown in Table 10, which determines from which distribution $\lambda_A$ and $\lambda_P$ are sampled. If the distance between the historical sample and the newly received sample is below a predefined distance threshold (set to 0.8 in our experiment), $\lambda_A, \lambda_P$ are sampled from a uniform distribution with a lower mean. If the distance between the historical sample and the newly received sample exceeds the predefined distance threshold, $\lambda_A, \lambda_P$ are sampled from a truncated Gaussian distribution with a higher mean and lower standard deviation.

## B.3 More experimental results

Due to space limitations in the main text, we added comparisons with more methods in the appendix. Tables 11 and 12 show comparisons with additional Test-time adaptation methods, including EVO-MSN (Qin et al., 2024) and TAFSA (Kim et al., 2025), under practical scenario settings. Tables 13 and 14 present comparisons with more series decomposition and Test-time adaptation methods under ideal scenario settings.

## B.4 VAE Models Architecture

We use the Total Correlation Variational Autoencoder (TC-VAE) (Chen et al., 2018) to compute the distance in latent space between the current incoming sample and the historical samples stored in the memory bank. The model is trained for 100 epochs with a learning rate of 3e-3, using the Evidence Lower Bound (ELBO) as the loss function, and a batch size of 128. The latent dimensions and the $\beta$ parameter are set to 10 and 5, respectively. Below, we provide detailed information about the encoder and decoder architectures, which differ across datasets due to variations in input channels.

Table 11: The MSE results of the online time-series forecasting task. We compare extensive competitive models under different prediction lengths following the setting of Proceed. The best results are in **bold**, and the second best are underlined.

| Method | | **BTOA** | Proceed | SOLID++ | OneNet | FsNet | CycleNet | EVO-MSN | TAFAS | Dish-TS | PatchTST | DER++ | ER |
|---|---|---|---|---|---|---|---|---|---|---|---|---|---|
| ETTh1 | 24 | **0.708** | 0.729 | 0.745 | 0.780 | 0.993 | 0.753 | 0.769 | 0.743 | 0.761 | 0.756 | 0.834 | 0.811 |
| | 48 | **0.806** | 0.886 | 0.848 | 0.896 | 1.089 | 0.857 | 0.881 | 0.851 | 0.882 | 0.887 | 0.921 | 0.901 |
| | 96 | **0.930** | 1.003 | 0.977 | 1.025 | 1.359 | 0.986 | 1.010 | 0.979 | 1.074 | 1.086 | 1.036 | 1.019 |
| ETTh2 | 24 | **1.688** | 1.801 | 2.021 | 2.606 | 2.941 | 2.228 | 2.435 | 1.990 | 1.931 | 1.893 | 2.790 | 2.492 |
| | 48 | **2.772** | 3.291 | 3.442 | 3.921 | 4.090 | 3.701 | 3.777 | 3.426 | 3.391 | 3.283 | 4.090 | 3.799 |
| | 96 | **4.880** | 5.790 | 5.802 | 6.248 | 6.245 | 6.674 | 6.112 | 5.808 | 6.054 | 5.976 | 6.363 | 6.022 |
| ETTm1 | 24 | 0.454 | **0.422** | 0.455 | 0.766 | 0.627 | 0.604 | 0.677 | 0.451 | 0.459 | 0.470 | 0.775 | 0.745 |
| | 48 | 0.592 | 0.579 | **0.578** | 0.978 | 0.855 | 0.808 | 0.858 | 0.579 | 0.608 | 0.598 | 0.847 | 0.817 |
| | 96 | **0.657** | 0.660 | 0.659 | 0.882 | 1.348 | 0.876 | 0.811 | 0.659 | 0.697 | 0.673 | 0.887 | 0.859 |
| ETTm2 | 24 | **0.614** | 0.617 | 0.699 | 0.768 | 0.877 | 0.895 | 0.747 | 0.690 | 0.693 | 0.658 | 1.838 | 1.626 |
| | 48 | **1.020** | 1.090 | 1.113 | 1.202 | 1.261 | 1.431 | 1.173 | 1.117 | 1.118 | 1.063 | 2.223 | 1.989 |
| | 96 | **1.556** | 1.979 | 1.822 | 3.102 | 4.224 | 2.121 | 2.718 | 1.837 | 2.041 | 1.867 | 2.733 | 2.505 |
| WTH | 24 | **0.712** | 0.728 | 0.735 | 0.774 | 0.877 | 0.874 | 0.762 | 0.734 | 0.741 | 0.732 | 1.502 | 1.444 |
| | 48 | 0.976 | **0.973** | 0.980 | 1.047 | 1.328 | 1.19 | 1.029 | 0.979 | 0.989 | 0.981 | 1.658 | 1.605 |
| | 96 | **1.261** | 1.264 | 1.263 | 1.320 | 1.714 | 1.471 | 1.309 | 1.261 | 1.269 | 1.263 | 1.793 | 1.750 |
| ECL | 24 | **3.941** | 3.978 | 4.156 | 4.112 | 6.194 | 4.343 | 4.122 | 4.182 | 4.432 | 4.143 | 11.877 | 11.304 |
| | 48 | **4.656** | 4.664 | 4.780 | 4.750 | 5.186 | 5.186 | 4.759 | 4.764 | 4.901 | 4.762 | 12.683 | 12.076 |
| | 96 | 5.675 | **5.672** | 5.835 | 5.703 | 12.851 | 6.350 | 5.726 | 5.887 | 6.791 | 5.791 | 13.221 | 12.671 |
| Traffic | 24 | **0.332** | 0.335 | 0.376 | 0.351 | 0.452 | 0.371 | 0.355 | 0.379 | 0.371 | 0.376 | 0.461 | 0.477 |
| | 48 | **0.356** | 0.358 | 0.378 | 0.374 | 0.498 | 0.395 | 0.372 | 0.376 | 0.393 | 0.380 | 0.501 | 0.519 |
| | 96 | **0.371** | 0.375 | 0.397 | 0.386 | 0.565 | 0.41 | 0.383 | 0.398 | 0.405 | 0.398 | 0.573 | 0.584 |

Table 12: The MAE results of the online time-series forecasting task. We compare extensive competitive models under different prediction lengths following the setting of Proceed. The best results are in **bold**, and the second best are underlined.

| Method | | **BTOA** | Proceed | SOLID++ | OneNet | FsNet | CycleNet | EVO-MSN | TAFAS | Dish-TS | PatchTST | DER++ | ER |
|---|---|---|---|---|---|---|---|---|---|---|---|---|---|
| ETTh1 | 24 | **0.533** | 0.534 | 0.552 | 0.559 | 0.624 | 0.556 | 0.557 | 0.550 | 0.761 | 0.552 | 0.604 | 0.593 |
| | 48 | **0.576** | 0.593 | 0.593 | 0.600 | 0.664 | 0.596 | 0.598 | 0.593 | 0.882 | 0.601 | 0.635 | 0.627 |
| | 96 | **0.625** | 0.650 | 0.645 | 0.648 | 0.752 | 0.644 | 0.647 | 0.646 | 1.074 | 0.664 | 0.675 | 0.668 |
| ETTh2 | 24 | **0.563** | 0.603 | 0.609 | 0.638 | 0.696 | 0.636 | 0.629 | 0.608 | 1.931 | 0.608 | 0.714 | 0.684 |
| | 48 | **0.666** | 0.733 | 0.718 | 0.729 | 0.797 | 0.752 | 0.726 | 0.720 | 3.391 | 0.720 | 0.801 | 0.778 |
| | 96 | **0.800** | 0.847 | 0.868 | 0.927 | 0.945 | 0.920 | 0.909 | 0.883 | 6.054 | 0.887 | 0.925 | 0.905 |
| ETTm1 | 24 | 0.408 | **0.393** | 0.406 | 0.487 | 0.484 | 0.481 | 0.463 | 0.413 | 0.459 | 0.415 | 0.579 | 0.566 |
| | 48 | 0.477 | 0.462 | **0.461** | 0.582 | 0.560 | 0.564 | 0.546 | 0.473 | 0.608 | 0.474 | 0.605 | 0.592 |
| | 96 | 0.517 | 0.519 | **0.503** | 0.586 | 0.606 | 0.594 | 0.561 | 0.511 | 0.697 | 0.510 | 0.619 | 0.608 |
| ETTm2 | 24 | **0.406** | 0.409 | 0.420 | 0.435 | 0.449 | 0.442 | 0.431 | 0.414 | 0.693 | 0.415 | 0.619 | 0.589 |
| | 48 | 0.515 | 0.521 | 0.490 | 0.511 | 0.519 | 0.516 | 0.505 | 0.490 | 1.118 | **0.486** | 0.658 | 0.629 |
| | 96 | 0.577 | 0.586 | **0.563** | 0.621 | 0.636 | 0.585 | 0.604 | 0.567 | 2.041 | 0.565 | 0.700 | 0.674 |
| WTH | 24 | **0.473** | 0.477 | 0.482 | 0.497 | 0.557 | 0.560 | 0.493 | 0.482 | 0.989 | 0.482 | 0.739 | 0.719 |
| | 48 | **0.473** | 0.477 | 0.482 | 0.497 | 0.557 | 0.560 | 0.493 | 0.482 | 0.989 | 0.482 | 0.739 | 0.719 |
| | 96 | **0.584** | 0.592 | 0.597 | 0.603 | 0.679 | 0.662 | 0.601 | 0.592 | 1.269 | 0.592 | 0.780 | 0.764 |
| ECL | 24 | **0.281** | 0.285 | 0.294 | 0.293 | 0.381 | 0.299 | 0.293 | 0.293 | 4.432 | 0.294 | 0.583 | 0.566 |
| | 48 | **0.309** | 0.312 | 0.315 | 0.313 | 0.435 | 0.322 | 0.314 | 0.315 | 4.901 | 0.315 | 0.600 | 0.583 |
| | 96 | 0.348 | 0.340 | 0.340 | **0.336** | 0.464 | 0.349 | 0.337 | 0.341 | 6.791 | 0.341 | 0.601 | 0.585 |
| Traffic | 24 | **0.231** | 0.232 | 0.276 | 0.250 | 0.316 | 0.264 | 0.258 | 0.272 | 0.371 | 0.276 | 0.322 | 0.331 |
| | 48 | **0.243** | 0.245 | 0.244 | 0.262 | 0.337 | 0.278 | 0.257 | 0.265 | 0.393 | 0.267 | 0.347 | 0.353 |
| | 96 | **0.248** | 0.249 | 0.249 | 0.263 | 0.371 | 0.284 | 0.259 | 0.275 | 0.405 | 0.278 | 0.372 | 0.376 |

Table 13: The MSE results of the online time-series forecasting task. We compare extensive competitive models under different prediction lengths following the setting of FsNet. * means 'Former'. The best results are in **bold**, and the second best are underlined.

| Method | | BTOA | OneNet | FsNet | TAFAS | iTrans* | CycleNet | Dish-TS | Revin | Stationary | Auto* | NLinear | Peri-mid* | PatchTST | DER++ | ER |
|---|---|---|---|---|---|---|---|---|---|---|---|---|---|---|---|---|
| ETTh1 | 1 | **0.223** | 0.235 | 0.286 | 0.229 | 0.223 | 0.389 | 0.257 | 0.237 | 0.383 | 0.501 | 0.287 | 0.445 | 0.246 | 0.239 | 0.240 |
| | 24 | **0.271** | 0.400 | 0.411 | 0.336 | 0.703 | 0.959 | 0.692 | 0.671 | 0.758 | 1.496 | 0.759 | 1.228 | 0.810 | 0.648 | 0.673 |
| | 48 | **0.265** | 0.447 | 0.402 | 0.356 | 0.828 | 1.080 | 0.941 | 0.792 | 0.747 | 0.891 | 0.846 | 0.986 | 0.831 | 0.606 | 0.634 |
| ETTh2 | 1 | 0.390 | 0.383 | 0.467 | 0.387 | 0.418 | 0.559 | 0.514 | 0.382 | 0.770 | 1.890 | 0.820 | 1.225 | **0.362** | 0.508 | 0.508 |
| | 24 | **0.505** | 0.538 | 0.693 | 0.522 | 1.716 | 2.206 | 1.584 | 1.741 | 2.088 | 3.010 | 2.332 | 2.608 | 1.622 | 0.828 | 0.808 |
| | 48 | **0.587** | 0.604 | 0.867 | 0.596 | 2.781 | 3.254 | 2.119 | 2.762 | 2.938 | 3.946 | 3.405 | 3.600 | 2.716 | 1.157 | 1.136 |
| ETTm1 | 1 | 0.106 | 0.117 | **0.104** | 0.112 | 0.106 | 0.125 | 0.120 | 0.122 | 0.111 | 0.385 | 0.109 | 0.255 | 0.116 | 0.110 | 0.114 |
| | 24 | **0.114** | 0.134 | 0.137 | 0.124 | 0.777 | 1.663 | 0.816 | 1.531 | 0.536 | 1.885 | 0.923 | 1.774 | 0.427 | 0.196 | 0.202 |
| | 48 | **0.118** | 0.118 | 0.124 | 0.118 | 0.783 | 1.648 | 1.322 | 1.018 | 1.433 | 1.991 | 0.804 | 1.820 | 0.553 | 0.208 | 0.220 |
| ETTm2 | 1 | 0.174 | 0.191 | 0.179 | 0.183 | **0.168** | 0.173 | 0.321 | 0.173 | 0.194 | 0.511 | 0.178 | 0.342 | 0.184 | 0.190 | 0.191 |
| | 24 | **0.206** | 0.267 | 0.233 | 0.237 | 0.639 | 0.659 | 0.611 | 0.652 | 0.954 | 1.075 | 0.695 | 0.867 | 0.547 | 0.307 | 0.310 |
| | 48 | **0.204** | 0.273 | 0.299 | 0.239 | 0.987 | 1.067 | 0.906 | 1.083 | 1.209 | 1.405 | 0.968 | 1.236 | 0.608 | 0.329 | 0.331 |
| WTH | 1 | 0.156 | 0.158 | 0.161 | 0.157 | 0.160 | 0.169 | 0.156 | 0.165 | **0.152** | 0.208 | 0.160 | 0.189 | 0.162 | 0.208 | 0.180 |
| | 24 | **0.161** | 0.189 | 0.189 | 0.175 | 0.375 | 0.388 | 0.340 | 0.370 | 0.428 | 0.445 | 0.350 | 0.417 | 0.372 | 0.270 | 0.293 |
| | 48 | **0.173** | 0.197 | 0.223 | 0.185 | 0.472 | 0.478 | 0.412 | 0.453 | 0.487 | 0.524 | 0.430 | 0.501 | 0.465 | 0.294 | 0.297 |
| ECL | 1 | 2.430 | 2.590 | 3.317 | 2.510 | **1.897** | 2.274 | 3.000 | 3.873 | 2.613 | 3.847 | 3.532 | 3.061 | 2.022 | 2.657 | 2.579 |
| | 24 | **2.493** | 2.700 | 6.071 | 2.597 | 4.009 | 6.585 | 4.006 | 4.862 | 3.469 | 7.289 | 7.148 | 6.937 | 4.325 | 8.996 | 9.327 |
| | 48 | **2.423** | 3.261 | 7.234 | 2.842 | 4.787 | 7.276 | 4.479 | 6.583 | 4.987 | 9.004 | 8.658 | 8.140 | 5.030 | 9.009 | 9.685 |
| Traffic | 1 | **0.232** | 0.233 | 0.295 | 0.233 | 0.298 | 0.313 | 0.410 | 0.257 | 0.418 | 1.052 | 0.904 | 0.683 | 0.533 | 0.280 | 0.286 |
| | 24 | **0.310** | 0.348 | 0.360 | 0.329 | 1.097 | 1.187 | 1.006 | 1.097 | 1.275 | 1.755 | 1.641 | 1.471 | 0.913 | 0.384 | 0.383 |
| | 48 | **0.351** | 0.384 | 0.378 | 0.368 | 1.615 | 1.786 | 1.479 | 1.678 | 1.765 | 2.281 | 2.182 | 2.034 | 1.519 | 0.398 | 0.394 |

Table 14: The MAE results of the online time-series forecasting task. We compare extensive competitive models under different prediction lengths following the setting of FsNet. * means 'Former'. The best results are in **bold**, and the second best are underlined.

| Method | | BTOA | OneNet | FsNet | TAFAS | iTrans* | CycleNet | Dish-TS | Revin | Stationary | Auto* | NLinear | Per-mid* | PatchTST | DER++ | ER |
|---|---|---|---|---|---|---|---|---|---|---|---|---|---|---|---|---|
| ETTh1 | 1 | 0.301 | 0.393 | 0.343 | 0.347 | **0.294** | 0.385 | 0.318 | 0.304 | 0.395 | 0.474 | 0.341 | 0.430 | 0.311 | 0.305 | 0.316 |
| | 24 | **0.325** | 0.442 | 0.436 | 0.384 | 0.524 | 0.621 | 0.517 | 0.510 | 0.565 | 0.801 | 0.559 | 0.711 | 0.570 | 0.534 | 0.547 |
| | 48 | **0.356** | 0.454 | 0.452 | 0.405 | 0.570 | 0.661 | 0.622 | 0.557 | 0.570 | 0.617 | 0.580 | 0.639 | 0.575 | 0.525 | 0.538 |
| ETTh2 | 1 | 0.362 | 0.351 | 0.371 | 0.357 | 0.352 | 0.384 | 0.362 | **0.344** | 0.383 | 0.574 | 0.426 | 0.479 | 0.351 | 0.375 | 0.376 |
| | 24 | **0.397** | 0.414 | 0.473 | 0.406 | 0.587 | 0.602 | 0.594 | 0.581 | 0.659 | 0.706 | 0.685 | 0.654 | 0.577 | 0.540 | 0.543 |
| | 48 | **0.436** | 0.445 | 0.516 | 0.441 | 0.676 | 0.692 | 0.677 | 0.664 | 0.722 | 0.772 | 0.796 | 0.732 | 0.672 | 0.577 | 0.571 |
| ETTm1 | 1 | 0.187 | 0.202 | 0.187 | 0.195 | 0.192 | 0.210 | 0.204 | 0.208 | 0.197 | 0.406 | 0.193 | 0.308 | **0.186** | 0.192 | 0.197 |
| | 24 | **0.222** | 0.243 | 0.249 | 0.233 | 0.529 | 0.692 | 0.535 | 0.704 | 0.449 | 0.842 | 0.566 | 0.767 | 0.471 | 0.326 | 0.333 |
| | 48 | **0.118** | 0.118 | 0.124 | 0.118 | 0.783 | 1.648 | 1.322 | 1.018 | 1.433 | 1.991 | 0.804 | 1.820 | 0.553 | 0.208 | 0.220 |
| ETTm2 | 1 | 0.226 | 0.233 | 0.229 | 0.230 | **0.221** | 0.224 | 0.271 | 0.226 | 0.228 | 0.373 | 0.221 | 0.299 | 0.228 | 0.231 | 0.233 |
| | 24 | **0.206** | 0.267 | 0.233 | 0.237 | 0.639 | 0.659 | 0.611 | 0.652 | 0.954 | 1.075 | 0.695 | 0.867 | 0.547 | 0.307 | 0.310 |
| | 48 | **0.204** | 0.273 | 0.299 | 0.239 | 0.987 | 1.067 | 0.906 | 1.083 | 1.209 | 1.405 | 0.968 | 1.236 | 0.608 | 0.329 | 0.331 |
| WTH | 1 | 0.197 | 0.201 | 0.215 | 0.199 | 0.205 | 0.210 | **0.195** | 0.211 | 0.196 | 0.287 | 0.203 | 0.249 | 0.200 | 0.235 | 0.244 |
| | 24 | **0.241** | 0.273 | 0.276 | 0.257 | 0.399 | 0.406 | 0.382 | 0.394 | 0.446 | 0.471 | 0.386 | 0.439 | 0.393 | 0.351 | 0.356 |
| | 48 | **0.255** | 0.278 | 0.303 | 0.267 | 0.467 | 0.468 | 0.440 | 0.452 | 0.484 | 0.514 | 0.450 | 0.491 | 0.459 | 0.359 | 0.363 |
| ECL | 1 | 0.266 | 0.258 | 0.542 | 0.262 | **0.218** | 0.229 | 0.315 | 0.331 | 0.508 | 0.335 | 0.314 | 0.282 | 0.341 | 0.421 | 0.506 |
| | 24 | **0.313** | 0.346 | 1.024 | 0.356 | 0.366 | 0.368 | 0.508 | 0.332 | 0.579 | 0.455 | 0.438 | 0.412 | 0.375 | 1.035 | 1.057 |
| | 48 | 0.462 | **0.400** | 1.089 | 0.431 | 0.346 | 0.393 | 0.583 | 0.379 | 0.789 | 0.635 | 0.587 | 0.514 | 0.399 | 1.048 | 1.074 |
| Traffic | 1 | 0.205 | 0.210 | 0.253 | 0.215 | 0.321 | 0.318 | 0.315 | 0.295 | 0.325 | 0.466 | 0.436 | 0.392 | 0.307 | 0.241 | 0.247 |
| | 24 | **0.261** | 0.269 | 0.265 | 0.287 | 0.519 | 0.498 | 0.508 | 0.502 | 0.575 | 0.584 | 0.567 | 0.541 | 0.508 | 0.289 | 0.299 |
| | 48 | 0.293 | 0.302 | **0.297** | 0.298 | 0.568 | 0.570 | 0.583 | 0.573 | 0.617 | 0.640 | 0.626 | 0.605 | 0.571 | 0.295 | 0.307 |

Table 15: Encoder Network for ETT dataset

| Layer Name | Output size | # of kernels | Kernel size | Stride | Activation |
|---|---|---|---|---|---|
| Input | $N \times 1 \times 60 \times 7$ | | | | |
| Convolution | $N \times 32 \times 26 \times 5$ | 32 | $9 \times 3$ | $2 \times 1$ | ReLU |
| Convolution | $N \times 32 \times 10 \times 3$ | 32 | $7 \times 3$ | $2 \times 1$ | ReLU |
| Convolution | $N \times 64 \times 2 \times 1$ | 64 | $5 \times 3$ | $3 \times 1$ | ReLU |
| Convolution | $N \times 128 \times 1 \times 1$ | 128 | $2 \times 1$ | $1 \times 1$ | ReLU |
| Convolution | $N \times 128 \times 1 \times 1$ | 128 | $1 \times 1$ | $1 \times 1$ | |

Table 16: Decoder Network for ETT dataset

| Layer Name | Output size | # of kernels | Kernel size | Stride | Activation |
|---|---|---|---|---|---|
| Input | $N \times 10 \times 1 \times 1$ | | | | |
| Transposed Convolution | $N \times 512 \times 2 \times 7$ | 512 | $2 \times 7$ | $1 \times 1$ | ReLU |
| Transposed Convolution | $N \times 128 \times 2 \times 7$ | 128 | $4 \times 1$ | $6 \times 1$ | ReLU |
| Transposed Convolution | $N \times 64 \times 16 \times 7$ | 64 | $4 \times 1$ | $2 \times 1$ | ReLU |
| Transposed Convolution | $N \times 32 \times 32 \times 7$ | 32 | $4 \times 1$ | $2 \times 1$ | ReLU |
| Transposed Convolution | $N \times 32 \times 1 \times 7$ | 1 | $4 \times 1$ | $2 \times 1$ | |

Table 17: Encoder Network for WTH dataset

| Layer Name | Output size | # of kernels | Kernel size | Stride | Activation |
|---|---|---|---|---|---|
| Input | $N \times 1 \times 60 \times 12$ | | | | |
| Convolution | $N \times 32 \times 18 \times 11$ | 32 | $9 \times 2$ | $3 \times 1$ | ReLU |
| Convolution | $N \times 32 \times 6 \times 5$ | 32 | $3 \times 3$ | $3 \times 2$ | ReLU |
| Convolution | $N \times 64 \times 2 \times 5$ | 64 | $3 \times 3$ | $3 \times 2$ | ReLU |
| Convolution | $N \times 128 \times 1 \times 1$ | 128 | $2 \times 2$ | $2 \times 1$ | ReLU |
| Convolution | $N \times 128 \times 1 \times 1$ | 128 | $1 \times 1$ | $1 \times 1$ | |

Table 18: Decoder Network for WTH dataset

| Layer Name | Output size | # of kernels | Kernel size | Stride | Activation |
|---|---|---|---|---|---|
| Input | $N \times 10 \times 1 \times 1$ | | | | |
| Transposed Convolution | $N \times 512 \times 2 \times 12$ | 512 | $2 \times 7$ | $1 \times 1$ | ReLU |
| Transposed Convolution | $N \times 128 \times 2 \times 12$ | 128 | $4 \times 1$ | $6 \times 1$ | ReLU |
| Transposed Convolution | $N \times 64 \times 16 \times 12$ | 64 | $4 \times 1$ | $2 \times 1$ | ReLU |
| Transposed Convolution | $N \times 32 \times 32 \times 12$ | 32 | $4 \times 1$ | $2 \times 1$ | ReLU |
| Transposed Convolution | $N \times 32 \times 1 \times 12$ | 1 | $4 \times 1$ | $2 \times 1$ | |

Table 19: Encoder Network for ECL dataset

| Layer Name | Output size | # of kernels | Kernel size | Stride | Activation |
|---|---|---|---|---|---|
| Input | $N \times 1 \times 60 \times 321$ | | | | |
| Convolution | $N \times 32 \times 18 \times 64$ | 32 | $9 \times 2$ | $3 \times 5$ | ReLU |
| Convolution | $N \times 32 \times 6 \times 13$ | 32 | $3 \times 3$ | $3 \times 5$ | ReLU |
| Convolution | $N \times 64 \times 2 \times 3$ | 64 | $3 \times 3$ | $3 \times 5$ | ReLU |
| Convolution | $N \times 128 \times 1 \times 1$ | 128 | $2 \times 2$ | $2 \times 3$ | ReLU |
| Convolution | $N \times 128 \times 1 \times 1$ | 128 | $1 \times 1$ | $1 \times 1$ | |

Table 20: Decoder Network for ECL dataset

| Layer Name | Output size | # of kernels | Kernel size | Stride | Activation |
|---|---|---|---|---|---|
| Input | $N \times 10 \times 1 \times 1$ | | | | |
| Transposed Convolution | $N \times 512 \times 2 \times 321$ | 512 | $2 \times 321$ | $1 \times 1$ | ReLU |
| Transposed Convolution | $N \times 128 \times 2 \times 321$ | 128 | $4 \times 1$ | $6 \times 1$ | ReLU |
| Transposed Convolution | $N \times 64 \times 16 \times 321$ | 64 | $4 \times 1$ | $2 \times 1$ | ReLU |
| Transposed Convolution | $N \times 32 \times 32 \times 321$ | 32 | $4 \times 1$ | $2 \times 1$ | ReLU |
| Transposed Convolution | $N \times 32 \times 1 \times 321$ | 1 | $4 \times 1$ | $2 \times 1$ | |

Table 21: Encoder Network for traffic dataset

| Layer Name | Output size | # of kernels | Kernel size | Stride | Activation |
|---|---|---|---|---|---|
| Input | $N \times 1 \times 60 \times 862$ | | | | |
| Convolution | $N \times 32 \times 18 \times 173$ | 32 | $9 \times 2$ | $3 \times 5$ | ReLU |
| Convolution | $N \times 32 \times 6 \times 35$ | 32 | $3 \times 3$ | $3 \times 5$ | ReLU |
| Convolution | $N \times 64 \times 2 \times 7$ | 64 | $3 \times 3$ | $3 \times 5$ | ReLU |
| Convolution | $N \times 128 \times 1 \times 1$ | 128 | $2 \times 2$ | $2 \times 5$ | ReLU |
| Convolution | $N \times 128 \times 1 \times 1$ | 128 | $1 \times 1$ | $1 \times 1$ | |

Table 22: Decoder Network for traffic dataset

| Layer Name | Output size | # of kernels | Kernel size | Stride | Activation |
|---|---|---|---|---|---|
| Input | $N \times 10 \times 1 \times 1$ | | | | |
| Transposed Convolution | $N \times 512 \times 2 \times 862$ | 512 | $2 \times 862$ | $1 \times 1$ | ReLU |
| Transposed Convolution | $N \times 128 \times 2 \times 862$ | 128 | $4 \times 1$ | $6 \times 1$ | ReLU |
| Transposed Convolution | $N \times 64 \times 16 \times 862$ | 64 | $4 \times 1$ | $2 \times 1$ | ReLU |
| Transposed Convolution | $N \times 32 \times 32 \times 862$ | 32 | $4 \times 1$ | $2 \times 1$ | ReLU |
| Transposed Convolution | $N \times 32 \times 1 \times 862$ | 1 | $4 \times 1$ | $2 \times 1$ | |

