# OpenReview forum: "Batch Training for Streaming Time Series: A Transferable Augmentation Framework to Combat Distribution Shifts"
_TMLR — Accepted by TMLR_

### Review · Reviewer_fMvx · 2025-03-22

**Summary Of Contributions:**

This paper investigates the problem of test-time adaptation for time series forecasting. A simple and general method is proposed to select and augment similar historical samples for online batch learning, effectively reducing the noise introduced by distribution shift in the single test sample. Enhanced with a two-stream forecaster architecture, the proposed framework achieves competitive performance in the experimental results.

**Audience:**

No

**Broader Impact Concerns:**

The paper does not include a Broader Impact Statement. However, given that the proposed method is general-purpose and does not directly raise ethical or societal concerns, I believe this omission is acceptable.

**Claims And Evidence:**

No

**Requested Changes:**

1. **Please carefully review the related literature listed above**. Several recent works have addressed test-time or online adaptation in time series forecasting. A more comprehensive **literature review** is necessary, and **these papers should be included as baselines** in the experimental comparison to ensure fairness and relevance.

2. **Revise the experimental setup to follow the protocol discussed in [4].** While an exact replication is not required, the current setup suffers from overlapping prediction horizons during training, which makes the evaluation unreliable and unrealistic. This issue has been pointed out in [4]. As it stands, only the results for prediction length = 1 are valid under this setting.

3. Provide supporting evidence for key claims:

   * **Naive online learning vs. batched online learning**:
    The current claim that batch-based online learning improves performance over using a single sample per step is intuitive but lacks empirical support. Experiment results in [4] show that naive online learning with a single sample can sometimes outperform batch-based methods (Solid++). It would strengthen the paper to include a comparative analysis to verify this assumption.

    * **Explanation of sudden loss increases in Figure 2:**
    The paper attributes these spikes to significant distribution shifts in the data. However, this claim is currently speculative. Please include visualizations of the corresponding time series to substantiate this observation.

4. Minor issues:

     * The notations for ground truth and prediction in Section 3.3 appear to be swapped and are inconsistent with the definitions provided earlier in Section 3
    * Are the augmented samples use the original label for training? Additionally, how many augmented samples are generated per mini-batch?
    * The proposed framework incorporates two forecasters.  But the impact of this design is not investigated in ablation study.
     * The paper does not discuss the computational cost or training/inference time, which is an important consideration for online scenarios.

**Strengths And Weaknesses:**

### Strengths
1. The paper is clearly written and well-structured, making it easy to read and understand.
2. The problem of test-time adaptation is a timely and important direction to explore.
3. The proposed method is simple, general, and effective.
4. The experimental results of the proposed method are competitive in their setup.

### Weakness
1. The paper does not follow the latest developments in this area. In fact, several recent works have explored test-time adaptation for online time series forecasting [1-4]. However, this paper fails to include these recent advances and only compares with baselines that were published several years ago.
2. Due to the issue above, the paper still adopts the outdated online learning setup introduced by FSNet. This setup has been shown to be problematic in [4], as it overlooks the lagging nature of future horizons, which can result in label leakage during online training.
3. As a result of the concerns above, the experimental results and conclusions are difficult to evaluate fairly. While such a paper might have been acceptable two years ago, it does not meet the expectations for timeliness and relevance in 2025.
4. Some claims in the paper are made without sufficient evidence and come across as intuitive speculation rather than being supported by empirical or theoretical justification.

[1] Calibration of Time-Series Forecasting: Detecting and Adapting Context-Driven Distribution Shift (KDD 2024)

[2] Evolving Multi-Scale Normalization for Time Series Forecasting under Distribution Shifts (Openreview 2024)

[3] Battling the Non-stationarity in Time Series Forecasting via Test-time Adaptation (AAAI 2025)

[4] Proactive Model Adaptation Against Concept Drift for Online Time Series Forecasting (KDD 2025)

---

> ### Author Response · Authors · 2025-05-21
> **Rebuttal by Authors**
>
> We thank you for your valuable feedback and positive comments. We appreciate your thorough review and are glad that you found the BTOA module simple, efficient and plug and play!
>
> ## W1 & R1：About the comparison of added and updated baselines
>
> In Section 2, we provide a more comprehensive literature review. **In Table 2 of Section 4, we add comparisons with more advanced baselines, including CycleNet [1], Dish-TS [2], SOLID [3], and Proceed [4]**. In Table 11 and 12 of Appendix B.3, we include more complete experimental results, adding comparative experiments with EVO-MSN [5] and TAFAS [6]. **The experimental results demonstrate the effectiveness of the BTOA.**
>
> ## W2 & W3 & R2：About the use of more reasonable experimental settings
>
> In Section 4, we re-adopt the experimental setup of Proceed [4] and re-conduct performance comparison experiments as well as an Ablation Study on method modules under the practical experimental setup. Table 2 demonstrates the effectiveness of the method in practical scenarios. **Through comprehensive experiments in two experimental scenarios, we show that BTOA's superiority in both classical ideal experimental scenarios and practical scenarios outperforms the current state-of-the-art (SOTA), proving the effectiveness of BTOA.** We have updated the code at https://anonymous.4open.science/r/BTOA-310B.
>
> ## W4 & R3：Regarding Batch Online Learning Setting and Sudden Change in Loss Curve
>
> In Table 2 of Section 4, we add performance comparisons with Proceed and SOLID++. **Our method outperforms both Proceed and SOLID++**, demonstrating that batch-based online learning offers advantages over updating with individual samples at each step. The key lies in our method's ability to better leverage historical sample information from data distributions and reasonably perform data augmentation tailored to the special properties of time series, preserving critical frequency-domain information. Additionally, in Figure 4, we visualize the performance comparisons of PatchTST, Proceed, and our method under distribution shifts. **As shown, compared with PatchTST and Proceed, BTOA exhibits better capability to fit the data and resist the impact of distribution shifts.**
>
> In Figure 2 of Section 4, we visualize the distribution information of 200 samples before and after the loss increase peak. **As shown in Figure 2, the sample distributions clearly shifted before and after the peak, which empirically corroborates our inference that the model’s performance deteriorates sharply after a distribution shift.**
>
> ## R4: About the Processing of Augmented Samples
>
> Under our experimental setup, augmented samples are also trained using the original ground truth values. In Table 4, we add a parameter sensitivity experiment on training with samples of different batch sizes. In the experiment, we uniformly set the batch size to 16.
>
> ## R5: About the ablation study on the use of dual-stream forecaster
>
> We design ablation study targeting the two-stream forecaster. **Table 9 presents a comparison of the predictor architecture without adopting the two-stream paradigm.** **The experimental results demonstrate that the adoption of the two-stream forecaster exhibits significant effectiveness.**
>
> ## R6: About Computational Cost and Inference Latency
>
> In Figure 4, we discuss computational cost and inference latency. On the ETTh2 dataset, **compared to Proceed, BTOA improves prediction performance by 18% with only a 9% increase in inference time**; compared to OneNet, **BTOA enhances prediction performance by 28% while reducing inference time by 35%.** This comparison further highlights the superiority of BTOA.
>
> ## R7: About Typographical Errors
>
> Thank you for your careful reading. We have corrected this typographical error in the new version.
>
>
>
> We hope our response and revision were able to address any remaining concerns. If our revision is still unclear or does not fully address some of your concerns, please feel free to reach out.
>
> [1] Lin S, Lin W, Hu X, et al. Cyclenet: enhancing time series forecasting through modeling periodic patterns. NeurIPS. 2024.
>
> [2] Fan W, Wang P, Wang D, et al. Dish-ts: a general paradigm for alleviating distribution shift in time series forecasting. AAAI. 2023.
>
> [3] Chen M, Shen L, Fu H, et al. Calibration of time-series forecasting: Detecting and adapting context-driven distribution shift. KDD. 2024.
>
> [4] Zhao L, Shen Y. Proactive Model Adaptation Against Concept Drift for Online Time Series Forecasting. KDD. 2025.
>
> [5] Qin D, Li Y, Chen W, et al. Evolving Multi-Scale Normalization for Time Series Forecasting under Distribution Shifts. Arxiv. 2024
>
> [6] Kim H G, Kim S, Mok J, et al. Battling the Non-stationarity in Time Series Forecasting via Test-time Adaptation. AAAI. 2025

---

> > ### Comment · Reviewer_fMvx · 2025-05-22
> > **Reply to reubuttal**
> >
> > Thank you for the reply. The revision has addressed most of the issues I previously raised.
> > However, there are still some typos in the manuscript. For example, in the paragraph titled “Convergence of different deep models,” it should reference Figure 2 instead of Figure 1. Please carefully proofread the manuscript and correct these errors.

---

> > > ### Author Response · Authors · 2025-05-22
> > > **Rebuttal by Authors**
> > >
> > > Thank you for your careful reading. **We have carefully proofread the manuscript and corrected the typos in the new version.** We hope our responses have addressed any remaining concerns. If our rebuttal is still unclear or does not fully address your concerns, please feel free to contact us.

---

### Review · Reviewer_Xfxf · 2025-04-14

**Summary Of Contributions:**

The authors present an online test-time adaptation method for time series forecasting to alleviate prominent distribution shift challenges in related offline neural network methods. More specifically, they propose a **batch training-based method with transformable online augmentations** (so-called BTOA), which first leverages the historical information by selecting historical samples that are similar to the test-time distribution (named Transferable Historical Sample Selection - THSS component), followed by an augmentation technique that enhances the selected samples by phase and amplitude in the frequency domain (called the Transferable Online Augmentation - TOA component). Finally, a prediction module based on decomposition and a two-stream forecaster is used to encode the temporal data dependencies. The proposed BTOA framework can be incorporated into any online time series (batch-based) forecasting framework to boost robustness in terms of shifts between train and test distributions.

**Audience:**

Yes

**Claims And Evidence:**

No

**Requested Changes:**

1. Based on **[W1]**, could the authors provide intuitive examples or scenarios where their method clearly outperforms conventional test-time baselines? This would help clarify the practical advantages of the approach and better highlight the challenges addressed by the work. The introduction section could be further adapted to intuitively motivate the core contribution of the presented method.

2. Based on **[W2]**, the related works section could be improved to contain relevant modeling techniques in terms of architectural design. Additionally, important basic methods in terms of mitigating distribution shifts are missing. Could the authors include these babies in discussions and experiments? Are performance comparisons against the proposed baselines still favorable for BTOA?

3. Concerning **[W3]**, could the authors improve the presentation and connection of their methodological contribution to well-defined cases of time series distribution shifts?

4. To address **[W4]**, it is advised that the authors justify the significance of performance improvements to support their main claims in terms of experimental evaluation.

**Strengths And Weaknesses:**

The main **strong points** of the work are the following:
- Prominent online time series forecasting methods update the model with each new sample. However, this can be problematic if the incoming data significantly deviates from the historical distribution or is noisy. In contrast, in BTOA, authors enable *batch-based training*.
- The authors in the current benchmark present improved performances with respect to **both offline and online methods** in terms of cumulative point-wise metrics. Several *ablation studies* accompany the experimental evaluation of the method’s main components along with *sensitivity analysis* on the batch size selection and the models' convergence.

The following **weak points** can be identified for the submission:
- **[W1 - Positioning of the Work \& Missing Related Works]:** The main contribution of this work over existing online methods remains somewhat unclear, focusing primarily on batch-based adaptation and alternative design choices in key components (e.g., augmentation strategy, prediction module). Additionally, the title is very general and does not adequately refer to the methods-specific characteristics.
- **[W2 - Missing Related Works and Baselines]:** The prediction module is based on a simple decomposition framework to extract patterns in both the frequency and time domains. However, a substantial body of related work employs similar components, which the authors do not adequately discuss [1,2,3,4]. Furthermore, since the paper compares to offline methods (e.g., iTransformer), it would be valuable to extend these comparisons to recent state-of-the-art offline forecasting models, particularly those leveraging decomposition-based neural architectures. Additionally, several offline methods have been particularly designed for alleviating distribution shifts in forecasting (see baselines in RevIN and [5]), but the authors only compare against Reversible Instance Normalization.
- **[W3 - Missing Definitions\& Presentation of Distribution Shift Examples]:** The paper lacks a clear presentation of distribution shift definitions and specific examples/cases (e.g., concept drift is presented in FSNet, OneNet), which would help readers better understand the specific challenges in time series forecasting. Additionally, the analysis of distribution shifts in the proposed datasets is limited to ADF tests. In contrast, related methods such as RevIN provide more comprehensive insights using, for instance, density plots to visualize train-test distribution discrepancies and demonstrate how their approach mitigates these shifts.
- **[W4 - Incremental Performances and Missing Details in Experimental Evaluation]:** The performance gains in cumulative MSE/MAE are generally modest for the proposed method, and their statistical significance is unclear due to the absence of standard deviations. It is also not specified whether the authors conduct multiple runs to assess variance. For instance, in Table 6, the authors fail to clearly show the adaptability of BTOA since combining it with other common methods results in minor improvements for several horizons and datasets.


[1] Wu, H., Xu, J., Wang, J., & Long, M. (2021). Autoformer: Decomposition transformers with auto-correlation for long-term series forecasting. Advances in neural information processing systems, 34, 22419-22430.

[2] Lin, S., Lin, W., Hu, X., Wu, W., Mo, R., & Zhong, H. (2024). Cyclenet: enhancing time series forecasting through modeling periodic patterns. Advances in Neural Information Processing Systems, 37, 106315-106345.

[3] Zeng, A., Chen, M., Zhang, L., & Xu, Q. (2023, June). Are transformers effective for time series forecasting?. In Proceedings of the AAAI conference on artificial intelligence (Vol. 37, No. 9, pp. 11121-11128).

[4] Wu, Q., Yao, G., Feng, Z., & Shuyuan, Y. (2024). Peri-midformer: Periodic pyramid transformer for time series analysis. Advances in Neural Information Processing Systems, 37, 13035-13073.

[5] Fan, W., Wang, P., Wang, D., Wang, D., Zhou, Y., & Fu, Y. (2023, June). Dish-ts: a general paradigm for alleviating distribution shift in time series forecasting. In Proceedings of the AAAI conference on artificial intelligence (Vol. 37, No. 6, pp. 7522-7529).

---

> ### Author Response · Authors · 2025-05-21
> **Rebuttal by Authors**
>
> We thank you for your valuable feedback and positive comments. We appreciate your thorough review!
>
> ## W1 & R1：About clarifying methodological contributions and visualized results of model superiority
>
> In the Section 1, we reiterate the contributions of the BTOA method. Firstly, **to fully leverage historical distribution information, we introduce the Transferable Historical Sample Selection (THSS) module** with theoretical guarantees to accurately select historical samples from the memory bank that are closest to the test-time distribution. **This overcomes the inefficiency of traditional online methods relying on random sampling or full-volume updates**, enabling intelligent activation and on-demand utilization of historical information. Secondly, **to address the issue of distribution shift, we propose the Transferable Online Augmentation (TOA) module**, which supports batch training while **avoiding the frequency-domain distortion caused by traditional time-domain data augmentation methods.** TOA performs decoupling in the frequency domain and applies two-stream augmentation to the selected samples from both amplitude and phase dimensions, fully preserving the amplitude-phase coupling information during the augmentation process. This approach uniquely maintains the critical frequency-domain characteristics essential for time series forecasting.
>
> In Section 4, Figure 3, we supplemented the visualized prediction results of PatchTST, Proceed [1], and our method BTOA after the occurrence of distribution shift. **As shown, compared with PatchTST and Proceed, BTOA demonstrates superior data fitting capability and the ability to resist the impact of distribution shift.**
>
> ## W2&R2：About the addition of related work and more baseline comparisons
>
> In Section 4, we redesigned the experiments and incorporated real-world scenario experimental setups. In Table 2 and Table 3, comparisons with more advanced baselines are added, including Proceed [1], CycleNet [2], Dish-TS [3], SOLID [4]. **The experimental results demonstrate the effectiveness of BTOA. Compared with the previous SOTA models Proceed and OneNet, our method BTOA reduced the MSE by 6% and 13%, respectively.** We have updated the code at https://anonymous.4open.science/r/BTOA-310B.
>
> In Table 13 and 14 of Appendix B.3, we have added more comprehensive experimental results, including comparisons with methods such as Autoformer[5], NLinear[6], and Peri-midformer[7]—these methods are based on decomposition frameworks to simplify temporal patterns. **The experimental results further demonstrate the effectiveness of the proposed method.**
>
> ## W3 &R3：About the definition of distribution shift and specific examples.
>
> In Section 3, **we provide a specific problem definition for distribution shift.**
>
> In Figure 2 of Section 4, we **visualized the sample distributions of 200 samples on each side of the loss curve peak.** It can be observed that there is a significant shift in the sample distributions before and after the peak, which corroborates our hypothesis that **the sudden increase in loss is caused by substantial distributional changes in the data.** As shown, when distribution shift occurs, **the MSE increment of BTOA is significantly smaller than all baselines, and it declines faster, thereby maintaining more stable performance.**
>
> ## W4&R4：About the generality of the method and experimental standard deviation
>
> In Table 3 of Section 4, we verified the generality of BTOA by incorporating the BTOA module into TCN and iTransformer. **The experimental results demonstrate the generality of BTOA..**
>
> In Table 8 of Appendix A, **we report model performance standard deviations across seven general datasets to validate performance improvement effectiveness.** To further assess method generality, we conducted triplicate experiments with three random seeds and added standard deviations for TCN and iTransformer backbones in Table 8, whose results further confirm the method's efficacy.
>
> We hope our response and revision were able to address any remaining concerns. If our revision is still unclear or does not fully address some of your concerns, please feel free to reach out.
>
> [1] Zhao L, Shen Y. Proactive Model Adaptation Against Concept Drift for Online Time Series Forecasting. KDD. 2025.
>
> [2] Lin S, et al. Cyclenet: enhancing time series forecasting through modeling periodic patterns. NeurIPS. 2024.
>
> [3] Fan W,et al. Dish-ts: a general paradigm for alleviating distribution shift in time series forecasting. AAAI. 2023.
>
> [4] Chen M,et al. Calibration of time-series forecasting: Detecting and adapting context-driven distribution shift. KDD. 2024.
>
> [5] Wu H, et al. Autoformer: Decomposition transformers with auto-correlation for long-term series forecasting. NeurIPS. 2022.
>
> [6] Zeng A, et al. Are transformers effective for time series forecasting? AAAI. 2023.
>
> [7] Wu Q, et al. Peri-midformer: Periodic pyramid transformer for time series analysis. NeurIPS. 2024

---

### Review · Reviewer_wLse · 2025-05-06

**Summary Of Contributions:**

The paper proposes a novel framework called Batch Training with Transferable Online Augmentation (BTOA) for addressing online test-time adaptation (TTA) in multivariate time series forecasting. Three key innovations are introduced: (1) a Transferable Historical Sample Selection (THSS) module leveraging a VAE to identify historically similar samples, (2) a Transferable Online Augmentation (TOA) module performing frequency-domain augmentation to preserve critical time-series features, and (3) a prediction block combining time series decomposition and a two-stream forecaster for complex pattern extraction.

**Audience:**

Yes

**Broader Impact Concerns:**

None.

**Claims And Evidence:**

Yes

**Requested Changes:**

Please see weaknesses.

**Strengths And Weaknesses:**

Strengths:
1. This paper addresses distribution shifts in time series via frequency-domain augmentation, which is innovative and rarely explored.
2. Extensive experiments on real-world datasets with ablation studies validating each module’s contribution.

Weaknesses:
1. The manuscript emphasizes the online nature of the proposed time series forecasting method. However, it lacks a comprehensive analysis of computational costs and inference latency, which are critical considerations for real-time applications.
2. A systematic comparison with state-of-the-art test-time adaptation techniques for time series forecasting is notably absent, which would help substantiate the claimed advancements of the proposed method (e.g., [1][2][3]).
3. The paper lacks ablation experiments examining the two-stream paradigm, which is a core design element in both the TOA module (amplitude/phase separation) and the forecasting architecture.

[1] Online test-time adaptation of spatial-temporal traffic flow forecasting.
[2] Augmented Contrastive Clustering with Uncertainty-Aware Prototyping for Time Series Test Time Adaptation.
[3] Battling the Non-stationarity in Time Series Forecasting via Test-time Adaptation.

---

> ### Author Response · Authors · 2025-05-21
> **Rebuttal by Authors**
>
> We thank you for your valuable feedback and positive comments. We appreciate your thorough review and are  glad that you think the BTOA module is innovative!
>
> ## W1:About Computational Cost and Inference Latency
>
> In Figure 4, we discuss computational cost and inference latency. On the ETTh2 dataset, **compared to Proceed, BTOA improves prediction performance by 18% with only a 9% increase in inference time**; compared to OneNet, **BTOA enhances prediction performance by 28% while reducing inference time by 35%.** This comparison further highlights the superiority of BTOA.
>
> ## W2：About Comparisons with More Advanced Baselines
>
> In Section 4, we redesign the experiments and incorporated real-world scenario experimental setups. In Table 2 and Table 3, comparisons with more advanced baselines are added, including CycleNet [1], Dish-TS [2], SOLID [3], and Proceed [4]. Additionally, in Table 13 and 14 of the Appendix B.3, we add more comprehensive experimental results, including comparisons with methods such as Autoformer [5], NLinear [6], and TAFAS [7]. **The experimental results demonstrate the effectiveness of BTOA. Compared with the previous SOTA models Proceed and OneNet, our method BTOA reduced the MSE by 6% and 13%, respectively.** We have updated the code at https://anonymous.4open.science/r/BTOA-310B.
>
> ## W3：About Ablation Study of the Two-Stream Paradigm
>
> We design ablation study targeting the two-stream paradigm. **For the TOA module, we composed a comparative experiment without adopting amplitude- and phase-based data augmentation methods in Table 6; Table 9 presents a comparison of the predictor architecture without adopting the two-stream paradigm.** The experimental results demonstrate that the adoption of the two-stream paradigm exhibits significant effectiveness.
>
> We hope our response and revision were able to address any remaining concerns. If our revision is still unclear or does not fully address some of your concerns, please feel free to reach out.
>
>
>
> [1] Lin S, Lin W, Hu X, et al. Cyclenet: enhancing time series forecasting through modeling periodic patterns. NeurIPS. 2024.
>
> [2] Fan W, Wang P, Wang D, et al. Dish-ts: a general paradigm for alleviating distribution shift in time series forecasting. AAAI. 2023.
>
> [3] Chen M, Shen L, Fu H, et al. Calibration of time-series forecasting: Detecting and adapting context-driven distribution shift. KDD. 2024.
>
> [4] Zhao L, Shen Y. Proactive Model Adaptation Against Concept Drift for Online Time Series Forecasting. KDD. 2025.
>
> [5] Wu H, Xu J, Wang J, et al. Autoformer: Decomposition transformers with auto-correlation for long-term series forecasting. NeurIPS. 2022.
>
> [6] Zeng A, Chen M, Zhang L, et al. Are transformers effective for time series forecasting? AAAI. 2023.
>
> [7] Kim H G, Kim S, Mok J, et al. Battling the Non-stationarity in Time Series Forecasting via Test-time Adaptation. AAAI. 2025.

---

### Author Response · Authors · 2025-05-21
**General Response**

We would like to express our sincere gratitude to all the reviewers for their valuable time and effort in reviewing our paper. In this general response, we summarize the key revisions made in the updated manuscript and aim to address each of the specific points raised in the individual rebuttals. **All modifications in the main text are highlighted in red.** The main changes include:

1、We revise the title to better highlight the specific content of the methodology.

2、We add the experimental conclusions to the abstract.

3、In Section 1, We re-induct and summarize the contributions of the methodology.

4、In Section 2, we incorporate a more comprehensive literature review.

5、In Section 3, we add the problem definition of distribution shift and corrected typos.

6、In Section 4, we add more advanced baselines, re-run the experiments based on more reasonable experimental settings, and present specific cases of distribution shift along with visualization results demonstrating the superiority of the methodology.

7、In Appendix A.3, we add the experimental standard deviations of the methodology.

8、In Appendix B.3, we add more comprehensive and complete experimental result tables.

---

### Decision · Action_Editor_GVx9 · 2025-06-05

**Recommendation:** Accept as is

**Audience:**

Yes

**Audience Explanation:**

The paper's findings would be interesting if it included ML methods for the time-series community, which has quite a significant representation in TMLR.

**Claims And Evidence:**

Yes

**Claims Explanation:**

The reviewers acknowledge that the authors have substantially improved the paper by addressing previous concerns. Revisions include clearer positioning relative to related work, better problem formulation (especially regarding distribution shift), enhanced experiments with relevant baselines, and added ablation studies and analysis. The contribution is now seen as strong enough for publication in TMLR.